# Alpha-satellite RNA transcripts are repressed by centromere–nucleolus associations

Leah Bury[1†], Brittania Moodie[1†], Jimmy Ly[1,2], Liliana S McKay[1], Karen HH Miga[3], Iain M Cheeseman[1,2]*

[1]Whitehead Institute for Biomedical Research, Cambridge, United States; [2]Department of Biology, Massachusetts Institute of Technology, Cambridge, United States; [3]UC Santa Cruz Genomics Institute, University of California, Santa Cruz, Santa Cruz, United States

**Abstract** Although originally thought to be silent chromosomal regions, centromeres are instead actively transcribed. However, the behavior and contributions of centromere-derived RNAs have remained unclear. Here, we used single-molecule fluorescence in-situ hybridization (smFISH) to detect alpha-satellite RNA transcripts in intact human cells. We find that alpha-satellite RNA-smFISH foci levels vary across cell lines and over the cell cycle, but do not remain associated with centromeres, displaying localization consistent with other long non-coding RNAs. Alpha-satellite expression occurs through RNA polymerase II-dependent transcription, but does not require established centromere or cell division components. Instead, our work implicates centromere–nucleolar interactions as repressing alpha-satellite expression. The fraction of nucleolar-localized centromeres inversely correlates with alpha-satellite transcripts levels across cell lines and transcript levels increase substantially when the nucleolus is disrupted. The control of alpha-satellite transcripts by centromere-nucleolar contacts provides a mechanism to modulate centromere transcription and chromatin dynamics across diverse cell states and conditions.

*For correspondence:
icheese@wi.mit.edu

[†]These authors contributed equally to this work

Competing interests: The authors declare that no competing interests exist.

## Introduction

Chromosome segregation requires the function of a macromolecular kinetochore structure to connect chromosomal DNA and spindle microtubule polymers. Kinetochores assemble at the centromere region of each chromosome. The position of centromeres is specified epigenetically by the presence of the histone H3-variant, CENP-A, such that specific DNA sequences are neither necessary nor sufficient for centromere function. (*McKinley and Cheeseman, 2016*). However, despite the lack of strict sequence requirements, centromere regions are typically characterized by repetitive DNA sequences, such as the alpha-satellite repeats found at human centromeres. Understanding centromere function requires knowledge of the centromere-localized protein components, as well as a clear understanding of the nature and dynamics of centromere chromatin. Although originally thought to be silent chromosome regions, centromeres are actively transcribed (*Perea-Resa and Blower, 2018*). Prior work has detected α-satellite transcription at centromere and pericentromere regions based on the localization of RNA polymerase II (*Bergmann et al., 2012*; *Chan et al., 2012*) and the production of centromere RNA transcripts (*Chan et al., 2012*; *Saffery et al., 2003*; *Wong et al., 2007*). Centromere transcription and the resulting RNA transcripts have been proposed to play diverse roles in kinetochore assembly and function (*Biscotti et al., 2015*; *Blower, 2016*; *Fachinetti et al., 2013*; *Ferri et al., 2009*; *Grenfell et al., 2016*; *Ideue et al., 2014*; *McNulty et al., 2017*; *Quénet and Dalal, 2014*; *Rošić and Erhardt, 2016*; *Wong et al., 2007*). However, due to limitations for analyses of centromere transcripts that average behaviors across

populations of cells and based on varying results between different studies, the nature, behavior, and contributions of centromere-derived RNAs have remained incompletely understood.

Here, we used single-molecule fluorescence in-situ hybridization (smFISH) to detect alpha-satellite RNA transcripts in individual, intact human cells. Our results define the parameters for the expression and localization of centromere and pericentromere-derived transcripts across a range of conditions. We find that the predominant factor controlling alpha-satellite transcription is the presence of centromere–nucleolar contacts, providing a mechanism to modulate centromere transcription and the underlying chromatin dynamics across diverse cell states and conditions.

## Results and discussion

### Quantitative detection of alpha-satellite RNAs by smFISH

Prior work analyzed centromere RNA transcripts primarily using population-based assays, such as RT-qPCR and RNA-seq, or detected centromere RNAs in spreads of mitotic chromosomes. To visualize alpha-satellite RNA transcripts in individual intact human cells, we utilized single-molecule fluorescence in-situ hybridization (smFISH), a strategy that has been used to detect mRNAs and cellular long non-coding RNAs (lncRNAs) (*Raj et al., 2008*). The high sensitivity of smFISH allows for the accurate characterization of number and spatial distribution of RNA transcripts.

Alpha-satellite DNA is degenerate such that it can vary substantially between different chromosomes with the presence of higher-order repeats of alpha-satellite variants (*Waye and Willard, 1987*; *Willard and Waye, 1987b*). Thus, we first designed targeted probe sets to detect RNAs derived from centromere regions across multiple chromosomes: (1) Sequences complementary to a pan-chromosomal consensus alpha-satellite sequence (labeled as 'ASAT'), (2) sequences that target supra-chromosomal family 1 (SF1) higher-order arrays, present on chromosomes 1, 3, 5, 6, 7, 10, 12, 16, and 19 (labeled as 'SF1') (*Alexandrov et al., 2001*; *Uralsky et al., 2019*), and (3) sequences that are enriched for transcripts from the Supra-Chromosomal family three higher-order arrays present on chromosomes 1, 11, 17 and X (labeled as 'SF3'), with an increased number of targets on chromosome 17 (D17Z1) (*Willard and Waye, 1987a*). Second, we designed probes that detect sequences enriched on specific chromosomes including the X chromosome (DXZ1, labeled 'X'; *Miga et al., 2014*; *Willard et al., 1983*) and chromosome 7 (D7Z2, labeled as '7.2'; *Waye et al., 1987*). For complete sequence information and an analysis of sequence matches to different chromosomes, see *Supplementary files 1* and *2*. Alpha-satellite DNA can span megabases of DNA on a chromosome, whereas the active centromere region is predicted to be as small as 100 kb in many cases (*McKinley and Cheeseman, 2016*). Thus, these smFISH probes will detect RNA transcripts from both the active centromere region and flanking pericentric alpha-satellite DNA.

In asynchronously cycling HeLa cells, we detected clear foci using smFISH probe sets for ASAT, SF1, and SF3 (*Figure 1A*). To ensure that this signal was not due to non-specific hybridization of the RNA probes to genomic DNA, we treated cells with RNase A prior to hybridization. The RNA- FISH signal was diminished substantially after RNase A treatment (*Figure 1A,B*), confirming the ribonucleic source of the observed signal. As an additional validation of these probes to confirm that they are recognizing alpha-satellite-derived sequences, we used them in a modified procedure to conduct DNA FISH. DNA FISH revealed multiple DNA-associated puncta that were distributed throughout the nucleus in interphase and aligned along the spindle axis on metaphase/anaphase chromosomes (*Figure 1—figure supplement 1A*), consistent with the behavior of centromere regions. In contrast to the ASAT, SF1, and SF3 probes, we did not detect smFISH foci using oligos designed to recognize transcripts derived from the centromere regions of chromosome seven or the X chromosome (*Figure 1—figure supplement 1B*). As the absence of signal could reflect a variety of technical features of probe design or a detection limit for the expression level or length of these sequences, we chose not to pursue these probes further. To quantify the number of distinct RNA-FISH foci, we used CellProfiler (*Carpenter et al., 2006*) to measure the number of foci per nucleus systematically using z-projections of the acquired images. The number of smFISH foci varied between individual cells, but averaged approximately four foci/cell for the ASAT, SF1, and SF3 probe sets in HeLa cells (*Figure 1C*).

Transcription of non-coding RNAs often occurs from both strands of DNA at a given locus. We therefore tested whether we could detect antisense (relative to the 'sense' probes used above)

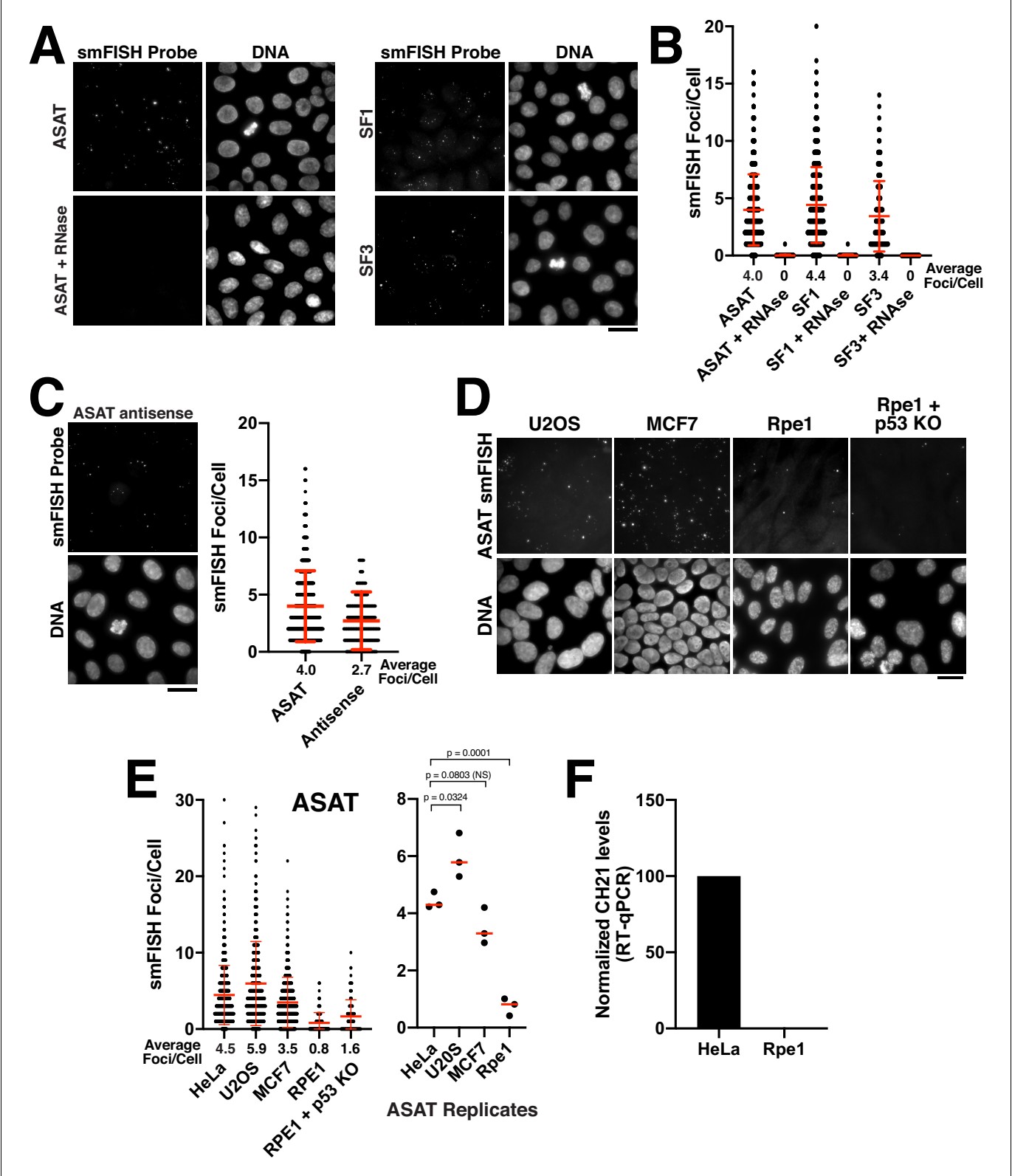

**Figure 1.** Quantitative detection of centromere RNAs using smFISH. (**A**) Detection of alpha-satellite RNA transcripts by smFISH in asynchronous HeLa cells. Designed probes detected RNAs derived from centromeres across subsets of multiple chromosomes, but with distinct specificity (see *Supplementary file 2*; ASAT, SF1, and SF3 repeats). Treatment of cells with RNase A prior to hybridization diminished RNA-smFISH signals. (**B**) Quantification of smFISH foci in the presence or absence of RNase A treatment indicates that the signal observed is due to a ribonucleic source. Points

*Figure 1 continued on next page*

*Figure 1 continued*

represent the number of foci per cell for each cell test. Error bars represent the mean and standard deviation of at least 100 cells. (**C**) Detection of anti-sense alpha-satellite transcripts in HeLa cells for the ASAT smFISH probe sequences. Error bars represent the mean and standard deviation of at least 100 cells. (**D**) Images showing varying abundance of alpha-satellite RNA across cell lines (based on smFISH foci), with RPE-1 cells displaying overall lower levels of centromere smFISH foci. For the RPE-1 + p53 KO condition, p53 was eliminated using an established TP53 iKO cell line (*McKinley and Cheeseman, 2017*). (**E**) Left, quantification indicating the variation of smFISH foci across selected cell lines. Error bars represent the mean and standard deviation of at least 100 cells. Right, average smFISH foci/cell for multiple independent replicates to enable statistical comparisons. p-values represent T-tests conducted on replicates of smFISH foci numbers for each selected cell line. (**F**) Graph showing quantification of RT-qPCR for alpha-satellite transcripts from chromosome 21. Levels of chromosome 21 alpha-satellite RNAs was not detected in Rpe1 cells and was therefore set to 0 in the figure. The levels of alpha-satellite transcripts in RPE-1 cells are reduced compared to HeLa cells. A semi-quantitative assessment of the RT-PCR data (with no standard curve interpolation, see *Figure 1—figure supplement 1D*) indicated a ~ 20-fold reduction in alpha-satellite transcripts in RPE-1 cells relative to HeLa. We performed three biological replicates of the RT-qPCR. Scale bars, 25 μm.

The online version of this article includes the following source data and figure supplement(s) for figure 1:

**Source data 1.** Source data for the RT-qPCR experiments shown in *Figure 1F* and *Figure 1—figure supplement 1* – panel D.

**Figure supplement 1.** Centromere RNA levels vary across cell lines.

alpha-satellite transcripts in Hela cells using smFISH. Indeed, for the ASAT probe sequences, we were able to visualize ~3 foci/cell using antisense smFISH probes, similar to numbers using the sense probe set (four foci/cell) (*Figure 1C*). Antisense transcription at the centromere has also been previously reported across a variety of species (*Carone et al., 2009*; *Choi et al., 2011*; *Chueh et al., 2009*; *Ideue et al., 2014*; *Koo et al., 2016*; *Li et al., 2008*; *May et al., 2005*).

The level of transcription for centromeric and pericentric satellite DNA has been proposed to vary between developmental stages and tissue types (*Maison et al., 2010*; *Pezer and Ugarković, 2008*). In addition, changes in centromere and pericentromere transcription have been observed in cancers (*Ting et al., 2011*). Therefore, we next sought to analyze differences in smFISH foci across different cell lines using the ASAT and SF1 probe sets. We selected the chromosomally-unstable osteosarcoma cell line U2OS, the breast cancer cell line MCF7, and the immortalized, but non-transformed hTERT-RPE-1 cell line. We found that the levels of alpha-satellite transcripts varied modestly across cell lines (*Figure 1D,E*; *Figure 1—figure supplement 1C*), with RPE-1 cells displaying overall lower levels of smFISH foci. As an additional confirmation of these behaviors, we tested the presence of alpha-satellite transcripts by RT-qPCR. Using a previously validated RT-qPCR primer pair against the alpha-satellite array on chromosome 21 (*Molina et al., 2016*; *Nakano et al., 2003*), we observed dramatically reduced levels of alpha-satellite transcripts in RPE-1 cells compared to HeLa cells (*Figure 1—figure supplement 1D*; *Figure 1F*). To test whether the transformation status of the cell line correlated with the level of smFISH foci, we eliminated the tumor suppressor p53 in RPE-1 cells using our previously-established inducible knockout strategy (*McKinley and Cheeseman, 2017*). Eliminating p53 did not substantially alter the levels of alpha-satellite smFISH foci in Rpe1 cells (*Figure 1D,E*) indicating that other factors likely contribute to the observed cellular levels of alpha-satellite RNA transcripts. Together, this strategy provides the ability to quantitatively detect centromere and peri-centromere-derived alpha-satellite RNA transcripts using smFISH probes against alpha-satellite sequences and demonstrates that human cell lines display varying levels of alpha-satellite transcripts.

## Analysis of alpha-satellite transcript localization and cell-cycle control

We next sought to assess the localization of alpha-satellite RNA transcripts within a cell. Prior work suggested that non-coding centromere transcripts are produced in cis and remain associated with the centromere from which they are derived, including through associations with centromere proteins (*McNulty et al., 2017*). Other studies support the action of centromere-derived RNAs in trans (*Blower, 2016*), but again acting at centromeres. To investigate the distribution of the centromere transcripts, we performed combined immunofluorescence and smFISH to visualize alpha-satellite transcripts relative to centromeres and microtubules. In interphase cells, smFISH foci localized within the nucleus (*Figure 2A*). Thus, unlike many mRNAs, alpha-satellite-derived RNAs are not exported to the cytoplasm. Although we detected colocalization of alpha-satellite RNAs with a subset of centromeres in HeLa cells, only ~10% of smFISH foci overlapped with centromeres (*Figure 2A,B*). In mitotic cells, smFISH foci did not associate with chromatin (*Figure 2C*). Instead, during all stages of

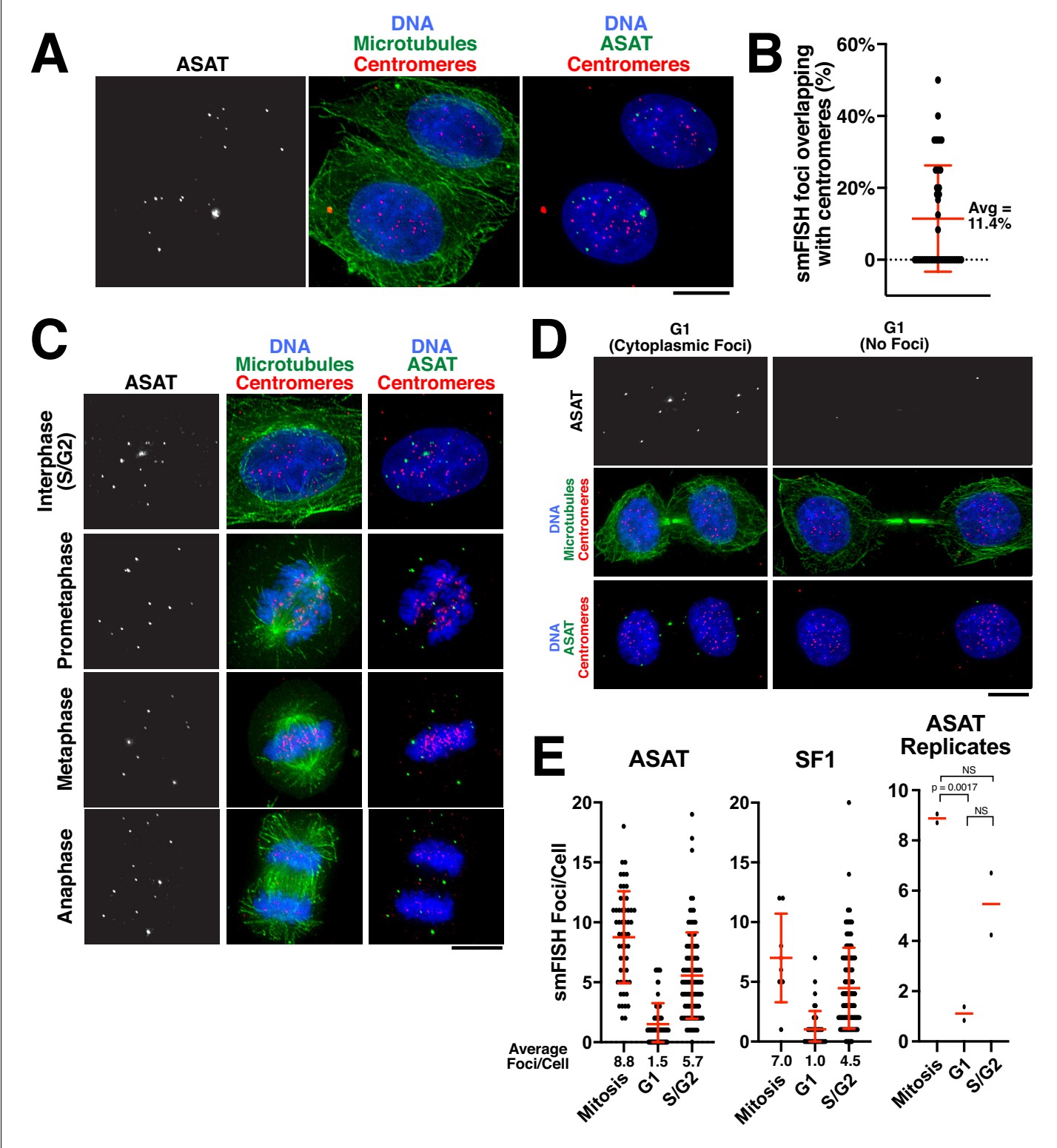

**Figure 2.** Analysis of centromere RNA foci across the cell cycle. (A) Immunofluorescence images (using anti-tubulin antibodies in green and anti-centromere antibodies (ACA) in red) showing alpha-satellite derived transcripts (smFISH; ASAT probe sets) localized to the nucleus during interphase in HeLa cells. The majority of detected transcripts do not co-localize with centromeres. (B) Graph showing the fraction of ASAT smFISH foci that overlap with centromeres by immunofluorescence. Each point represents one cell. n = 36 cells. (C) Immunofluorescence of HeLa cells (as in A) throughout the cell cycle reveals smFISH foci are separable from chromatin in mitosis. (D) Immunofluorescence-smFISH analysis indicates that progression of cells into

*Figure 2 continued on next page*

*Figure 2 continued*

G1 (defined by cells with a mid-body) results in the nuclear exclusion of smFISH foci. Left: Foci are located in the cytoplasm after the nuclear envelope reforms. Right: Foci are absent, possibly reflecting the degradation of cytoplasmic RNA. (E) Quantification of smFISH foci throughout the cell cycle (for either ASAT or SF1 probe sets) reveals that transcripts levels are high in S/G2 and mitotic cells, but reduced as cells exit mitosis into G1. A T-test was conducted on independent replicates of the ASAT smFISH data for each selected cell-cycle state. Error bars represent the mean and standard deviation of at least 8 cells/replicate. Scale bars, 10 μm.

mitosis, alpha-satellite RNA transcripts appeared broadly distributed within the cytoplasm. Finally, as the cells exited mitosis into G1, the smFISH foci remained distinct from the chromosomal DNA and were thus excluded from the nucleus when the nuclear envelope reformed (*Figure 2D*). Similar patterns of cell-cycle dependent localization changes with mitotic exclusion from chromatin have been reported for other cellular long non-coding RNAs (*Cabili et al., 2015*; *Clemson et al., 1996*). In contrast to our findings that alpha-satellite transcripts are primarily separable from centromere loci, prior work from others found close associations between alpha-satellite transcripts and centromeres (*Blower, 2016*; *Bobkov et al., 2018*; *McNulty et al., 2017*; *Rošić et al., 2014*). Based on these different behaviors, we hypothesize that the smFISH approach using the native fixation conditions detects mature alpha-satellite transcripts, but is unable to detect nascent RNAs in the process of transcription. Thus, once transcribed, alpha-satellite non-coding RNAs visualized by smFISH display nuclear localization, but are not tightly associated with the centromere regions from which they are derived.

We next analyzed the temporal changes in alpha-satellite transcript numbers during the cell cycle. In contrast to other genomic loci, RNA Polymerase II is present at human and murine centromeres during mitosis (*Chan and Wong, 2012*; *Perea-Resa et al., 2020*). In addition, centromere transcription during G1 has been proposed to play a role in CENP-A loading (*Bobkov et al., 2018*; *Chen et al., 2015*; *Quénet and Dalal, 2014*). Recent work measuring the levels of satellite transcripts originating from specific centromeres in human cells suggested the presence of stable RNA levels during the entire cell cycle (*McNulty et al., 2017*). smFISH provides the capacity to measure the levels of alpha-satellite transcripts in individual cells over the course of the cell cycle. We utilized combined immunofluorescence-smFISH to simultaneously label alpha-satellite RNA transcripts and microtubules, allowing us to distinguish between G1 cells (due to the presence of a mid-body), an S/G2 interphase population, and mitotic cells. In contrast to previous observations, our analysis revealed that the transcripts detected by our smFISH method increased in S/G2 and remained stable throughout mitosis (*Figure 2E*). We note that a G2/M peak of transcript levels has been reported for murine Minor Satellite transcripts (*Ferri et al., 2009*). However, as cells exited mitosis into G1, transcripts detected by smFISH were reduced (*Figure 2E*). We speculate that this may result from the nuclear exclusion of the existing alpha-satellite transcripts, which would make this more susceptible to degradation by cytoplasmic RNAses. Thus, alpha-satellite transcript levels fluctuate over the cell cycle with G1 as a period of low transcript numbers, either indicating reduced transcription during this cell-cycle stage or the increased elimination of alpha-satellite-derived RNA transcripts.

## Alpha-satellite RNAs are products of Pol II-mediated transcription

Previous studies have suggested that centromeres are actively transcribed by RNA polymerase II. RNA polymerase II localizes to centromeres in *S. pombe*, *Drosophila melanogaster*, and human cells, including at centromeric chromatin on human artificial chromosomes (HACs) and at neocentromeres (*Bergmann et al., 2011*; *Catania et al., 2015*; *Chan and Wong, 2012*; *Chueh et al., 2009*; *Ferri et al., 2009*; *Li et al., 2008*; *Ohkuni and Kitagawa, 2011*; *Perea-Resa et al., 2020*; *Quénet and Dalal, 2014*; *Rošić et al., 2014*; *Wong et al., 2007*). However, it remains possible that additional polymerases contribute to the transcription of alpha-satellite regions. To determine the polymerases that are responsible for generating the alpha-satellite transcripts detected by our smFISH assay, we treated Hela cells with small-molecule inhibitors against all three RNA polymerases. We found a significant reduction in alpha-satellite smFISH foci following inhibition of RNA Polymerase II activity using the small-molecule THZ1 (*Figure 3A–C*; *Figure 3—figure supplement 1A,B*), which targets the RNA Pol II activator Cdk7 (*Kwiatkowski et al., 2014*). In contrast, we did not detect a reduction in smFISH foci following treatment with inhibitors against RNA polymerase I (small-molecule inhibitor BHM-21; *Colis et al., 2014*) or RNA polymerase III (ML-60218; *Wu et al.,*

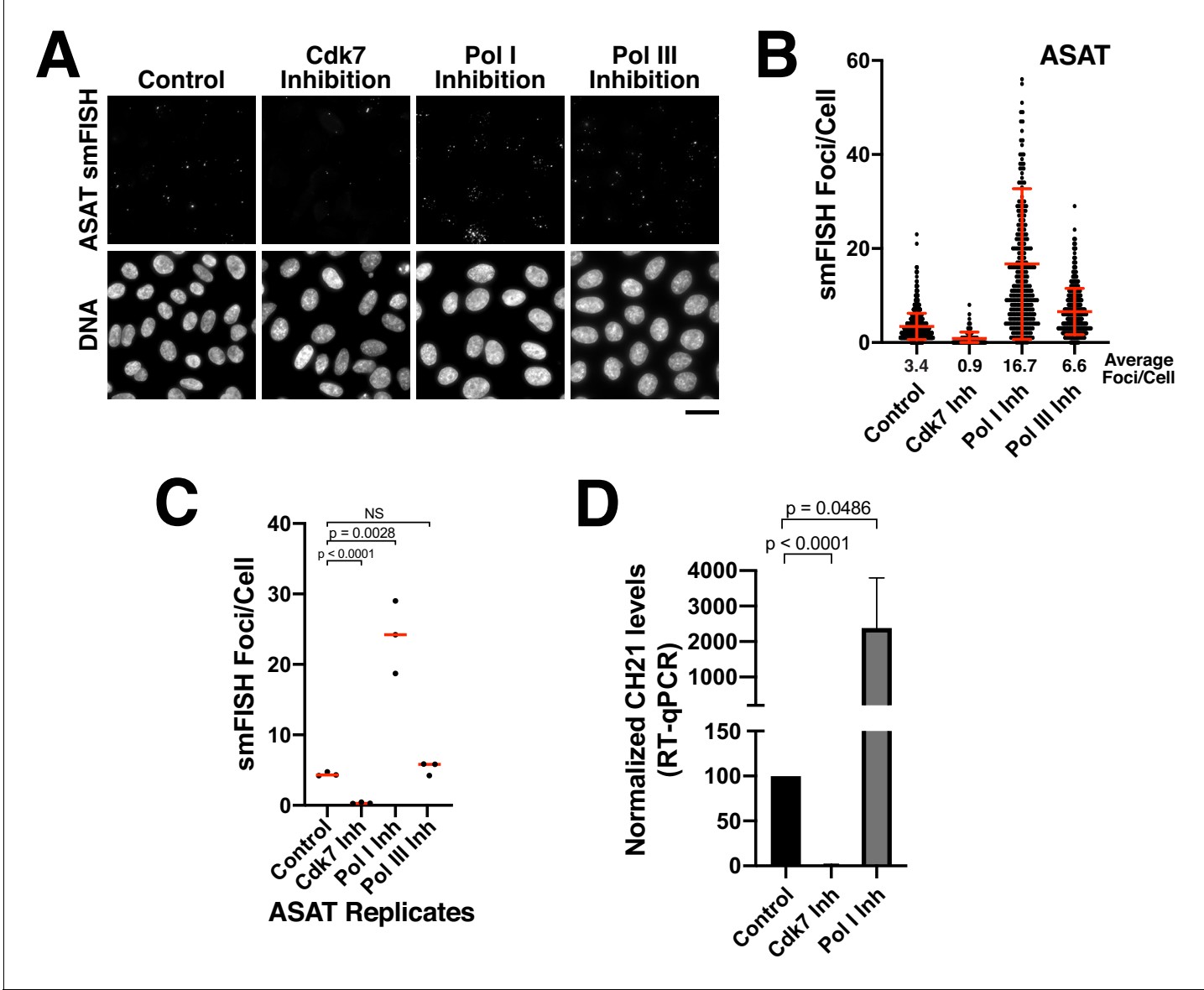

**Figure 3.** Alpha-satellite RNAs are products of Pol II-mediated transcription. (**A**) Treatment of HeLa cells with small-molecule inhibitors reveals that alpha-satellite transcripts are mediated by RNA polymerase II. Cells were treated with the RNA Polymerase I inhibitor BMH-21 (24 hr), the RNA Polymerase III inhibitor ML-60218 (24 hr), or the Cdk7 inhibitor THZ1 (5 hr), which inhibits RNA Polymerase II initiation. Transcripts were identified using the ASAT smFISH probe set. (**B**) Quantification of smFISH foci from (**A**) after treatment of HeLa cells with small-molecule inhibitors against Cdk7, RNA Pol I, and RNA Pol III. smFISH foci were substantially reduced after inhibition of RNA Pol II activator, Cdk7, but increased by RNA Pol I inhibition. Error bars represent the mean and standard deviation of at least 240 cells. (**C**) Graph showing independent replicates of ASAT smFISH foci for each small-molecule inhibitor treatment (Cdk7, RNA Pol I, and RNA Pol III). P-values represent T-tests for the indicated comparisons. (**D**) RT-qPCR quantification reveals significantly reduced levels of chromosome 21 alpha-satellite transcripts of cells treated by the Cdk7 inhibitor THZ1 for 5 hr, but increased levels following RNA polymerase I inhibition (24 hr treatment) when compared to control HeLa cells. The levels of alpha-satellite RNA from chromosome 21 detected was outside of our quantifiable range in cells treated with CDK7 inhibitor and thus was set to 0. The mean of 3 biological replicates was plotted and error bars represent the standard deviation. P-value represents the results of a T-test.

The online version of this article includes the following source data and figure supplement(s) for figure 3:

**Source data 1.** Source data for the RT-qPCR experiments shown in *Figure 3D*.
**Figure supplement 1.** Analysis of centromere RNAs following RNA polymerase inhibition.

*2003*; *Figure 3A–C*; *Figure 3—figure supplement 1A,B*). Instead, as discussed below, we found dramatically increased alpha-satellite smFISH foci following RNA polymerase I inhibition. Consistent with the effects of RNA polymerase I and II inhibition on alpha-satellite transcript levels as detected by smFISH, RT-qPCR analyses indicated substantially decreased chromosome 21 alpha-satellite transcripts following CDK7 inhibition, but increased levels following RNA polymerase I inhibition (*Figure 3D*). This indicates that the alpha-satellite RNA transcripts detected by smFISH are products of RNA Pol II-mediated transcription.

## Functional analysis of the protein requirements for alpha-satellite transcripts

We next sought to determine the requirements for the production of alpha-satellite transcripts. Centromere DNA functions as a platform for assembly of the kinetochore structure (*McKinley and Cheeseman, 2016*), an integrated scaffold of protein interactions that mediates the connection between the DNA and microtubules of the mitotic spindle. One possibility to explain the observed transcription of centromere regions, including at neocentromere loci lacking alpha-satellite sequences, is that centromere and kinetochore components act to recruit the RNA Polymerase machinery. To test this, we selectively eliminated diverse centromere and kinetochore components using a panel of CRISPR inducible knockout cell lines expressing dox-inducible Cas9 and guide RNAs (*McKinley and Cheeseman, 2017*; *McKinley et al., 2015*). We targeted the centromere-specific H3 variant CENP-A, the CENP-A chaperone HJURP (to block new CENP-A incorporation), the centromere alpha-satellite DNA binding protein CENP-B, the constitutive centromere components CENP-C, CENP-N, and CENP–W, and the outer kinetochore microtubule-binding protein Ndc80. Our prior work has documented the efficacy of each of these inducible knockout cell lines (*McKinley and Cheeseman, 2017*; *McKinley et al., 2015*). Consistently, we found that the gene targets were effectively eliminated from centromeres throughout the population for the CENP-A, CENP-B, and CENP-C inducible knockout cell lines (*Figure 4—figure supplement 1A*; also see *McKinley et al., 2015*). Eliminating these centromere and kinetochore components did not prevent the presence of alpha-satellite RNA-smFISH foci (*Figure 4A*). In contrast, the number of foci/cell increased in many of these inducible knockout cell lines, from moderate increases in most knockout cell lines to a substantial increase in CENP-C inducible knockout cells (*Figure 4A*). This suggests that centromere components are not required for the specific recruitment of RNA Polymerase II to centromere regions, although active centromeres may act to retain RNA Polymerase II during mitosis due to the persistence of sister chromatid cohesion (*Perea-Resa et al., 2020*).

We also tested the contribution of non-centromere-localized cell division components to alpha-satellite transcription. Because of its DNA-based nature, the centromere is subject to cell-cycle-specific challenges that include chromatin condensation, cohesion, and DNA replication. We thus sought to assess whether disruption of any of these complexes would influence alpha-satellite RNA transcript levels. To do this, we targeted proteins involved in centromere regulation (Sgo1 and BubR1), DNA replication (Mcm6, Gins1, Orc1, and Cdt1), sister chromatid cohesion (ESCO2, Scc1), chromosome condensation (Smc2, CAPG, CAPG2, TOP2A), and nucleosome remodeling (SSRP1). Strikingly, despite the diverse roles of these proteins in different aspects of centromere function, none of these inducible knockouts resulted in reduced levels of ASAT alpha-satellite transcripts as detected by smFISH analysis (*Figure 4—figure supplement 1B*). Instead, in many cases we detected a modest increase in alpha-satellite smFISH foci in the inducible knockout cells. Overall, our results indicate alpha-satellite transcription does not require the presence of specific DNA binding proteins, DNA structures, or cell division components, and instead that multiple factors act to restrict transcription at centromeres.

## CENP-C acts to repress alpha-satellite RNA levels

Of proteins that we tested, eliminating CENP-C had a particularly substantial effect on the number of smFISH foci (*Figure 4A*). To confirm this behavior following the loss of CENP-C, we repeated these experiments for both the ASAT and SF1 smFISH probes (*Figure 4B,C*; *Figure 4—figure supplement 1C*). In both cases, we observed a strong increase in smFISH foci. To test whether this behavior was specific to HeLa cells, we analyzed the CENP-C inducible knockout in RPE-1 cells. Although there are fewer ASAT smFISH foci in the parental RPE-1 cells, eliminating CENP-C resulted

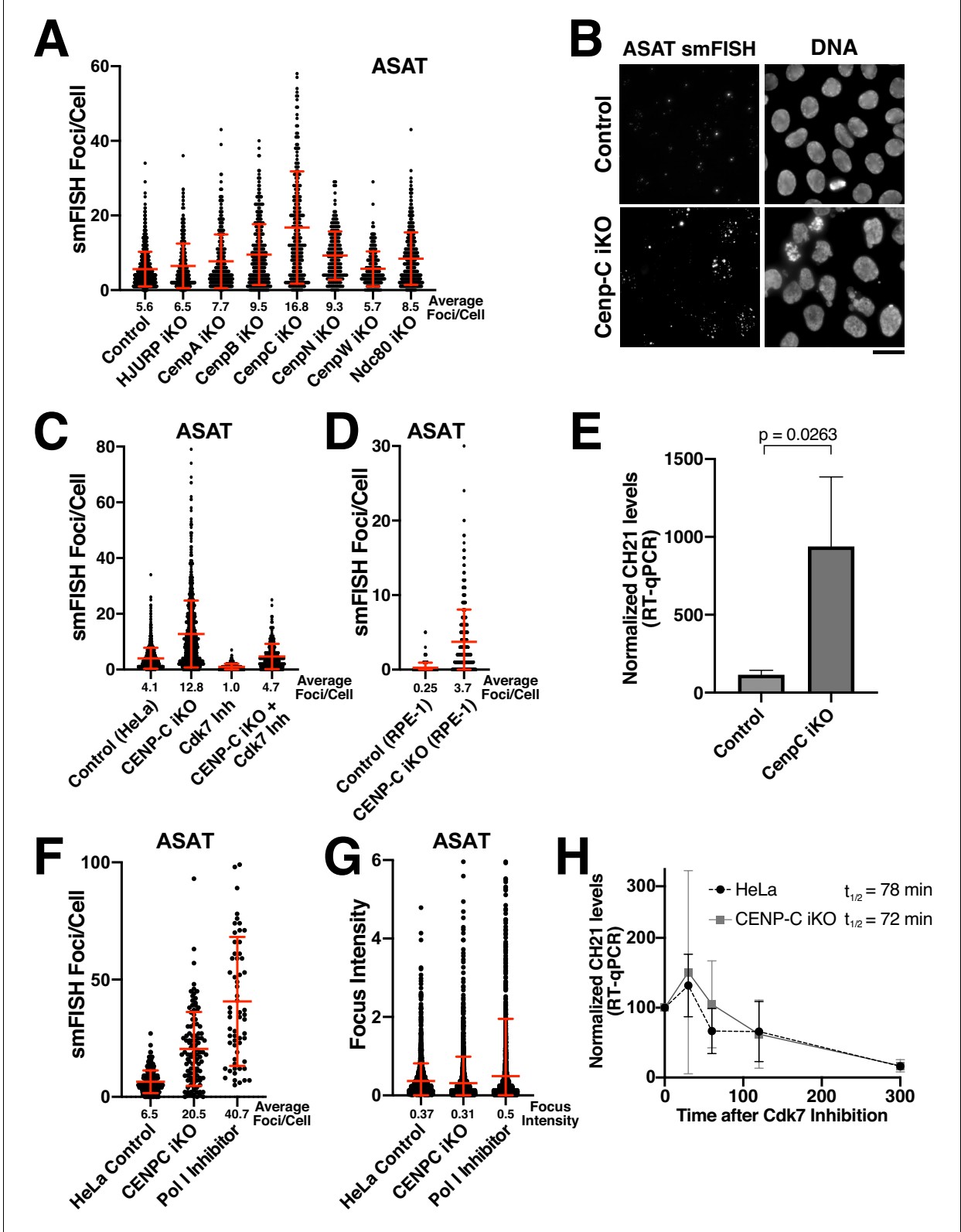

**Figure 4.** Eliminating CENP-C results in substantially increased alpha-satellite transcript numbers. (**A**) Quantification of smFISH foci (ASAT probe set) after elimination of selected centromere and kinetochore components reveals that centromere components are not required for the production of alpha-satellite transcripts. Inducible knockouts were generated using Cas9 using previously described cell lines (***McKinley and Cheeseman, 2017***; ***McKinley et al., 2015***). Notably, inducible knockout of CENP-C results in a substantial increase in smFISH foci. Error bars represent the mean and

*Figure 4 continued on next page*

*Figure 4 continued*

standard deviation of at least 240 cells. (B) Representative images showing the substantial increase in smFISH foci after elimination of the centromere component CENP-C. (C) Quantification of ASAT smFISH foci under the indicated conditions. The increase in alpha-satellite transcripts in cells depleted for CENP-C depends on RNA Polymerase II, as THZ1 treatment (Cdk7 inhibition; 5 hr) resulted in a substantial reduction in smFISH foci in both control cells and CENP-C inducible knockout cells. (D) Quantification of smFISH foci in CENP-C inducible knockout RPE-1 cells reveals that the increase in alpha-satellite transcripts following CENP-C knockout is not specific to HeLa cells. Error bars represent the mean and standard deviation of at least 170 cells. (E) RT-qPCR for alpha-satellite transcripts from chromosome 21 indicates a substantial increase in steady state alpha-satellite RNA levels in HeLa CENP-C inducible knockout cells. The mean of three biological replicates for control and four biological replicates for the CENP-C inducible knockouts was plotted. Error bars represent the standard deviation. P-value represents the results of a T-test. (F) Quantification of smFISH foci number in CENP-C inducible KO cells and Pol I-inhibited (24 hr treatment) cells compared to HeLa cell controls. (G) Quantification of the intensity of individual smFISH foci from the same experiment tested in F showing similar intensities despite the increase in foci number. (H) The half-life of alpha-satellite RNAs derived from chromosome 21 was determined in HeLa and CENP-C inducible knockout cells by RT-qPCR various times following RNA polymerase II inhibition (THZ1 treatment). The level of chromosome 21 alpha-satellite RNA was normalized to GAPDH, a stable mRNA. The half-life of these centromeric transcripts is 78 and 72 min in HeLa and CENP-C inducible knockout cells, respectively. Graph shows mean and standard deviation for two biological replicates. Scale bars, 25 µm.

The online version of this article includes the following source data and figure supplement(s) for figure 4:

**Source data 1.** Source data for the RT-qPCR experiments shown in *Figure 4D and H*.
**Figure supplement 1.** Analysis of alpha-satellite transcripts following disruption of cell division factors.

in a strong increase in the number of ASAT smFISH foci (*Figure 4D*). Moreover, we observed a substantial increase in steady state alpha-satellite RNA levels in HeLa CENP-C inducible knockout cells based on RT-qPCR (*Figure 4E*). We also note that recent work found that CENP-C overexpression resulted in decreased RNA Polymerase II occupancy at centromere regions (*Melters et al., 2019*). This increase in alpha-satellite transcripts in cells depleted for CENP-C depends on RNA Polymerase II, as THZ1 treatment resulted in a clear reduction in smFISH foci for the ASAT and SF1 probe sets in both control cells and CENP-C inducible knockout cells (*Figure 4C*; *Figure 4—figure supplement 1C*). The changes in smFISH foci that we observe in the CENP-C and RNA Polymerase I-inhibited cells likely reflects the number of independent and diffusible transcripts, as we did not detect a corresponding change in smFISH focus intensity (*Figure 4F,G*). Importantly, despite the increased numbers of smFISH foci in CENP-C inducible knockout cells, we found that alpha-satellite RNA transcripts displayed a similar half-life for their turnover based on RT-qPCR in control HeLa cells and CENP-C inducible knockout cells based on their loss following RNA Polymerase II inhibition (THZ1 treatment; *Figure 4H*). As the half-life of the alpha-satellite smFISH foci is similar in each case, this suggests that increased numbers of smFISH foci reflects increased transcription of alpha-satellite DNA instead of the increased stability of alpha-satellite transcripts. As eliminating CENP-C potently disrupts the localization of all centromere proteins (*McKinley et al., 2015*), this suggests that centromere and kinetochore formation could act as a physical block to restrict the passage of RNA polymerase through the centromere, downregulating alpha-satellite transcript levels. Alternatively, kinetochore proteins could act to create a repressive environment for transcription (see below).

## Centromere–nucleolar associations act to repress alpha-satellite transcript levels

In the functional analysis described above, we were surprised that most perturbations resulted in increased centromere smFISH RNA foci instead of a loss of signal. The largest increases were observed for the depletion of CENP-C (*Figure 4A–C*) and the inhibition of RNA Polymerase I (*Figure 3B*). RNA Polymerase I transcribes rDNA, but also has an important role in assembling the nucleolus, which creates a repressive transcriptional environment. Given reported connections between the centromere and nucleolus in prior work (*Ochs and Press, 1992*; *Padeken et al., 2013*; *Wong et al., 2007*), we hypothesized that alpha-satellite transcription occurs at a basal level, but that this transcription is repressed by associations between the centromere and the nucleolus. To test this model, we first visualized centromeres and nucleoli in human cells. In HeLa and RPE-1 cells, a subset of centromeres overlap with the nucleolus, as marked with antibodies against Ki-67 (*Figure 5A,B*) or Fibrillarin (*Figure 5—figure supplement 1A*). However, other centromeres are present outside of the nucleoli within the rest of the nucleus. This contrasts with work in *Drosophila* cells, where centromeres from all four chromosomes are found in close proximity surrounding the

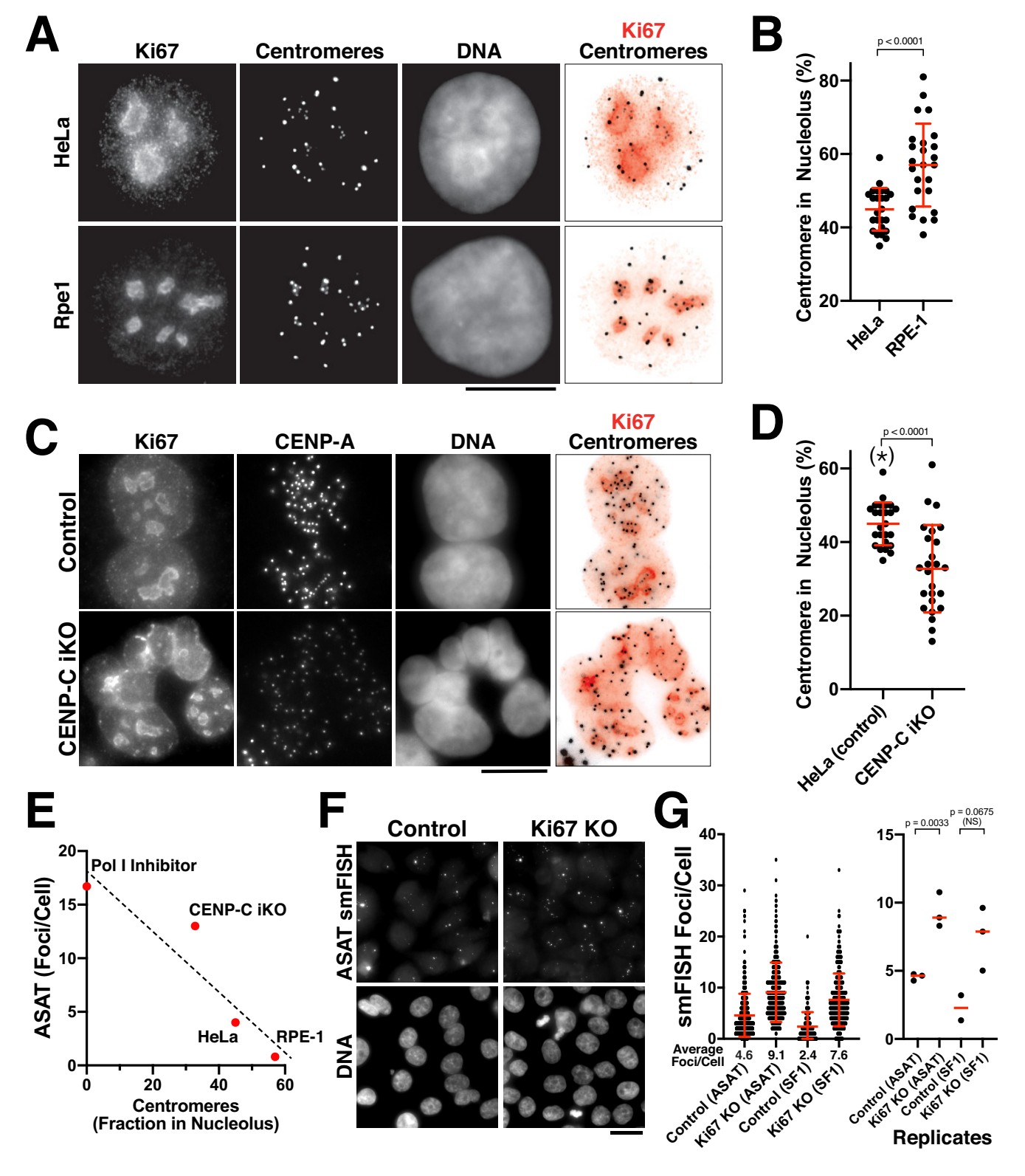

**Figure 5.** The nucleolus represses centromere RNA production. (**A**) Immunofluorescence of HeLa (Top) and RPE1 (Bottom) cells showing the colocalization of centromeres with the nucleolus, as marked with antibodies against Ki-67 and anti-centromere antibodies (ACA). Scale bars, 10 μm. (**B**) Quantification reveals RPE1 cells have a greater fraction of centromeres that overlap with nucleoli (57%) compared to HeLa cells (44.6%). Error bars represent the mean and standard deviation of 25 cells. (**C**) Immunofluorescence of HeLa control (top) and HeLa CENP-C iKO (bottom) cells showing the

*Figure 5 continued on next page*

*Figure 5 continued*

colocalization of centromeres with the nucleolus, as marked with antibodies against Ki-67 and CENP-A. Scale bar, 10 µm. (D) Quantification reveals that depletion of CENP-C results in a reduced fraction of nucleoli-localized centromeres (32.8%) compared to control cells (44.6%). The asterisk indicates that the data from control cells is repeated from (B). Error bars represent the mean and standard deviation of 25 cells. (E) Graph showing the relationship between the number of ASAT smFISH foci (summarized from data in *Figures 1–4*) and the fraction of nucleolar-localized centromeres in the indicated conditions. RNA Polymerase I inhibition should eliminate nucleolar function, and so is listed as '0' for nucleolar centromeres. Dashed line shows a linear fit trendline. (F) smFISH analysis reveals an increase of alpha-satellite transcripts in Ki67 knockout cells (right) when compared to control (left). Scale bar, 25 µm. (G) Quantification reveals a 2–3 fold increase in alpha-satellite transcript levels for both the ASAT and SF1 smFISH probes in Ki67 stable knockout cells. Error bars represent the mean and standard deviation of at least 100 cells. Right, graph showing replicates of the indicated data. P-values indicate T-tests for ASAT and SF1 replicates for Ki67 knockout cells compared to the corresponding control.

The online version of this article includes the following source data and figure supplement(s) for figure 5:

**Figure supplement 1.** Analysis of centromere-nucleolar contacts.

**Figure supplement 1—source data 1.** Source data for the RT-qPCR experiments shown in *Figure 5—figure supplement 1* – panel E.

nucleolus (*Padeken et al., 2013*). Importantly, we observed an inverse relationship between the fraction nucleoli-localized centromeres and the numbers of alpha-satellite smFISH foci. First, we observed an increased fraction of nucleoli-localized centromeres in RPE-1 cells compared to HeLa cells (*Figure 5A,B*), correlating with the reduced numbers of alpha-satellite smFISH foci in RPE-1 cells (*Figure 1F*). Similarly, we found that CENP-C inducible knockout cells displayed a reduced fraction of nucleoli-localized centromeres (*Figure 5C,D*), again correlating with the increased alpha-satellite smFISH foci in these cells (*Figure 4C*). In contrast, we did not detect a change in centromere-nucleolar associations in the CENP-B inducible knockout (*Figure 5—figure supplement 1B*), which does not substantially alter smFISH foci numbers (*Figure 4A*). When the different conditions affecting the nucleolus are compared, there is a clear inverse relationship between nucleolar-localized centromeres and the number of ASAT smFISH foci per cell (*Figure 5E*).

To assess the functional relationship between the nucleolus and alpha-satellite transcription, we generated inducible knockout cell lines for the nucleolar components Fibrillarin and Ki-67 using our established inducible Cas9 knockout system (*McKinley and Cheeseman, 2017*). Induction of these knockouts resulted in increased levels of alpha-satellite transcripts, particularly for Ki-67 (*Figure 5—figure supplement 1C*). Although Ki-67 plays important roles in nucleogenesis, mitotic chromosome structure, and transcription of cell-cycle targets (*Booth et al., 2014*; *Cuylen et al., 2016*; *Sobecki et al., 2016*; *Sun et al., 2017*), deletion of Ki-67 is not lethal (*Sobecki et al., 2016*). Therefore, we additionally generated a stable Ki67 knockout cell line in HeLa cells (*Figure 5—figure supplement 1D*). Ki-67 knockout cells proliferated normally, but displayed a 2–3 fold increase in alpha-satellite transcript levels for both the ASAT and SF1 smFISH probes (*Figure 5F,G*). However, we note that we did not detect a significant change in alpha-satellite transcript levels based on RT-qPCR (*Figure 5—figure supplement 1E*). The discrepancy between the smFISH and qPCR may represent differences between single molecule and bulk assays, or technical considerations as the smFISH probes recognize alpha-satellite RNAs derived from multiple chromosomes, whereas the RT-qPCR experiments detect alpha-satellite transcripts from only chromosome 21. Despite the increased alpha-satellite transcript levels detected by smFISH, we did not detect notable consequences to centromere protein levels (based on the localization of CENP-A; *Figure 5—figure supplement 1D,F*) or chromosome mis-segregation (not shown). The combination of these data supports a model in which a properly functioning nucleolus and nucleolar-centromere connections act to limit centromere and pericentromere transcript levels.

## Roles for alpha-satellite transcripts and centromere transcription

Together, this work defines the parameters for the production of alpha-satellite RNA transcripts and demonstrates that centromere-nucleolar connections act to restrict alpha-satellite transcription. The nature of the behavior that we observed for alpha-satellite smFISH foci, including the lack of persistent localization to centromeres or mitotic structures, is inconsistent with a direct, physical role for these transcripts in cell division processes. Instead, we propose that the process of ongoing transcription at centromeres itself, rather than the presence of alpha-satellite-derived RNAs, is an important feature of centromere biology. Active transcription could act to promote the dynamics of centromeric chromatin, resulting in the gradual turnover of DNA-bound proteins, including

nucleosomes. For example, prior work using artificial tethering of chromatin and transcription factors to chromosome regions has suggested that centromere function requires an intermediate level of transcription with strongly repressive or activating states incompatible with centromere function (*Molina et al., 2017*; *Molina et al., 2016*; *Nakano et al., 2008*). In addition, our recent work found that transcription was required to promote the gradual turnover of CENP-A nucleosomes in non-dividing cells, resulting in continued 'rejuvenation' of centromere proteins (*Swartz et al., 2019*). As part of that work, we also found a similar behavior for non-centromere chromatin, with transcription acting to drive the turnover of histone H3 on the chromosome arms (*Swartz et al., 2019*), suggesting that this may be a general feature of non-coding regions. Together, we propose that basal centromere transcription acts to promote the turnover of DNA-bound proteins, providing a mechanism to ensure refresh CENP-A chromatin. Importantly, changes in nuclear and nucleolar organization and in centromere-nucleolar associations across cell types, between cell states (including both dividing and quiescent cells), and in disease states has the potential to create consequential changes to centromere transcription and centromere protein dynamics.

# Materials and methods

## Key resources table

| Reagent type (species) or resource | Designation | Source or reference | Identifiers | Additional information |
|---|---|---|---|---|
| Cell line (*H. sapiens*) | HeLa | Don Cleveland lab (UCSD) | HeLa | Cell line maintained in the Cheeseman Lab |
| Cell line (*H. sapiens*) | U20S | Don Cleveland lab (UCSD) | U20S | Cell line maintained in the Cheeseman Lab initially received from the Cleveland lab |
| Cell line (*H. sapiens*) | MCF7 | American Type Culture Collection | MCF7 | Cell line maintained in the Cheeseman Lab initially received from American Type Culture Collection |
| Cell line (*H. sapiens*) | RPE1 | Prasad Jallepalli Lab (MSKCC) | RPE1 | Cell line maintained in the Cheeseman Lab initially received from Dr. Prasad Jallepalli |
| Cell line (*H. sapiens*) | cTT20 (inducible Cas9 in HeLa) | PMID:26698661 | cTT20 | Cell line maintained in the Cheeseman Lab initially generated by Tonia Tsinman |
| Cell line (*H. sapiens*) | cTT33 (inducible Cas9 in RPE1) | PMID:28216383 | cTT33.1 | Cell line maintained in the Cheeseman Lab initially generated by Tonia Tsinman |
| Cell line (*H. sapiens*) | CENP-C iKO (in HeLa/cTT20) | PMID:26698661 | cKM153 | Cell line maintained in the Cheeseman Lab initially generated by Dr. Kara McKinley |
| Cell line (*H. sapiens*) | CENP-B iKO (in HeLa/cTT20) | PMID:28216383 | cKMKO C1.1 | Cell line maintained in the Cheeseman Lab initially generated by Dr. Kara McKinley |
| Cell line (*H. sapiens*) | CENP-A iKO (in HeLa/cTT20) | PMID:28216383 | cKMKO B12.1 | Cell line maintained in the Cheeseman Lab initially generated by Dr. Kara McKinley |
| Cell line (*H. sapiens*) | CENPN iKO (in HeLa/cTT20) | PMID:26698661 | cKMKO | Cell line maintained in the Cheeseman Lab initially generated by Dr. Kara McKinley |
| Cell line (*H. sapiens*) | HJURP iKO (in HeLa/cTT20) | PMID:28216383 | cKMKO E4.1 | Cell line maintained in the Cheeseman Lab initially generated by Dr. Kara McKinley |
| Cell line (*H. sapiens*) | CENPW iKO (in HeLa/cTT20) | PMID:28216383 | cKMKO H3.3 | Cell line maintained in the Cheeseman Lab initially generated by Dr. Kara McKinley |

*Continued on next page*

*Continued*

| Reagent type (species) or resource | Designation | Source or reference | Identifiers | Additional information |
|---|---|---|---|---|
| Cell line (*H. sapiens*) | Ndc80 iKO (in HeLa/cTT20) | PMID:28216383 | cKMKO F11.1 | Cell line maintained in the Cheeseman Lab initially generated by Dr. Kara McKinley |
| Cell line (*H. sapiens*) | Fibrillarin iKO (in HeLa/cTT20) | This Paper | cBM002 | Cell line maintained in the Cheeseman Lab initially generated by Brittania Moodie |
| Cell line (*H. sapiens*) | Ki67 iKO (in HeLa/cTT20) | This Paper | cBM3.10 | Cell line maintained in the Cheeseman Lab initially generated by Brittania Moodie |
| Cell line (*H. sapiens*) | Sgo1 iKO (in HeLa/cTT20) | PMID:28216383 | cKMKO H1.1 | Cell line maintained in the Cheeseman Lab initially generated by Dr. Kara McKinley |
| Cell line (*H. sapiens*) | BubR1 iKO (in HeLa/cTT20) | PMID:28216383 | cKMKO A8.1 | Cell line maintained in the Cheeseman Lab initially generated by Dr. Kara McKinley |
| Cell line (*H. sapiens*) | Mcm6 iKO (in HeLa/cTT20) | PMID:28216383 | cKMKO E2.2 | Cell line maintained in the Cheeseman Lab initially generated by Dr. Kara McKinley |
| Cell line (*H. sapiens*) | Gins1 iKO (in HeLa/cTT20) | PMID:28216383 | cKMKO D11.1 | Cell line maintained in the Cheeseman Lab initially generated by Dr. Kara McKinley |
| Cell line (*H. sapiens*) | Orc1 iKO (in HeLa/cTT20) | PMID:28216383 | cKMKO G2.1 | Cell line maintained in the Cheeseman Lab initially generated by Dr. Kara McKinley |
| Cell line (*H. sapiens*) | Cdt1 iKO (in HeLa/cTT20) | PMID:28216383 | cKMKO B9.1 | Cell line maintained in the Cheeseman Lab initially generated by Dr. Kara McKinley |
| Cell line (*H. sapiens*) | ESCO2 iKO (in HeLa/cTT20) | PMID:28216383 | cKMKO C9.2 | Cell line maintained in the Cheeseman Lab initially generated by Dr. Kara McKinley |
| Cell line (*H. sapiens*) | Scc1 iKO (in HeLa/cTT20) | PMID:28216383 | cKMKO G11.1 | Cell line maintained in the Cheeseman Lab initially generated by Dr. Kara McKinley |
| Cell line (*H. sapiens*) | Smc2 iKO (in HeLa/cTT20) | PMID:28216383 | cKMKO H5.1 | Cell line maintained in the Cheeseman Lab initially generated by Dr. Kara McKinley |
| Cell line (*H. sapiens*) | CAPG iKO (in HeLa/cTT20) | PMID:28216383 | cKMKO E9.2 | Cell line maintained in the Cheeseman Lab initially generated by Dr. Kara McKinley |
| Cell line (*H. sapiens*) | CAPG2 iKO (in HeLa/cTT20) | PMID:28216383 | cKMKO E11.2 | Cell line maintained in the Cheeseman Lab initially generated by Dr. Kara McKinley |
| Cell line (*H. sapiens*) | TOP2A iKO (in HeLa/cTT20) | PMID:28216383 | cKMKO G11.2 | Cell line maintained in the Cheeseman Lab initially generated by Dr. Kara McKinley |
| Cell line (*H. sapiens*) | SSRP1 iKO (in HeLa/cTT20) | PMID:28216383 | cKMKO G10.2 | Cell line maintained in the Cheeseman Lab initially generated by Dr. Kara McKinley |
| Antibody | DM1a (anti-tubulin); mouse monoclonal | Sigma Aldrich | T9026-.2ML | (1:10000) |
| Antibody | ACA (anti-centromere antibodies; human auto-immune serum) | Antibodies, Inc | 15-234-0001 | (1:1000) |
| Antibody | Anti-CENP-A (mouse monoclonal) | Abcam | ab13939 | (1:1000) |

*Continued on next page*

*Continued*

| Reagent type (species) or resource | Designation | Source or reference | Identifiers | Additional information |
|---|---|---|---|---|
| Antibody | Anti-Ki67 (rabbit polyclonal) | Abcam | ab15580 | (1:100) |
| Antibody | Anti-Fibrillarin (rabbit polyclonal) | Abcam | ab5821 | (1:300) |
| Antibody | Anti-CENP-C (rabbit polyclonal) | Cheeseman Lab (Whitehead Institute) | N/A | (1:1000) |
| Antibody | Anti-CENP-B (rabbit polyclonal) | Abcam | ab25734 | (1:1000) |
| Commercial assay or kit | Custom Stellaris RNA FISH probes (Quasar570 or Quasar670) | Biosearch Technologies; PixelBiotech GmbH | N/A | Probe sequences may be found in *Supplementary file 2* |
| Commercial assay or kit | Custom HuluFISH probes (Atto565) | PixelBiotech GmbH | N/A | Probe sequences may be found in *Supplementary file 2* |
| Commercial assay or kit | Maxima First Strand cDNA Synthesis Kit for RT-qPCR | Life Technologies (Thermo Scientific) | K1671 | |
| Commercial assay or kit | SYBR Green PCR Master Mix | Thermo Fisher Scientifc | A25742 | |
| Chemical compound, drug | TRIzol Reagent (Tri Reagent solution) | Life Technologies | AM9738 | |
| Chemical compound, drug | RNase A | Qiagen | 19101 | 1:1000 |
| Chemical compound, drug | BMH-21; Pol I Inh | Millipore; Sigma Aldrich | 509911; SML1183 | 1 µM |
| Chemical compound, drug | ML-60218; Pol III Inh | Fisher Scientific (Thermo Fisher Scientific) | 557403 | 20 µM |
| Chemical compound, drug | THZ1; Cdk7 Inh | Fisher Scientific (Thermo Fisher Scientific) | 5323720001 | 1 µM |
| Chemical compound, drug | Formamide (Deionized) | Life Technologies | AM9342 | |
| Chemical compound, drug | Ribonucleoside vanadyl complexes | Sigma Aldrich | R3380-5ML | |

## Cell culture

All cells were grown in Dulbecco's Modified Eagle's Medium (DMEM) supplemented with 10% Fetal Bovine Serum, 100 units/mL penicillin, 100 units/mL streptomycin, and 2 mM L-glutamine (Complete Media) at 37°C with 5% $CO_2$. Cell lines represent established and ongoing cell lines used by the Cheeseman lab. They are validated based on their behavior and properties. All cell lines are tested for mycoplasma contamination on a regular and ongoing basis. For experiments using inducible knockout cell lines, cells were seeded onto uncoated glass coverslips and doxycycline (DOX, Sigma) was added at 1 mg/L for 48 hr. Cells were fixed and stained at 4 or 5 days following DOX addition. For inhibitor experiments, RNA Polymerase inhibitors were added to cells at the following concentrations: BMH-21 (RNAPI inhibitor; Millipore) at 1 µM; ML-60218 (RNAP III inhibitor; Fisher) at 20 µM; THZ1 (Cdk7 Inhibitor; Fisher Scientific) at 1 µM. Treatment times are indicated in the figure legends.

Inducible knockouts for nucleolar components in HeLa cells were created as described previously (*McKinley and Cheeseman, 2017*). Briefly, sgRNA sequences were cloned into pLenti-sgRNA (*Wang et al., 2015*), and used to generate lentiviruses for stable infection in cells harboring inducible Cas9 (HeLa cells – cTT20; RPE-1 cell - cTT33). Cells were then selected with puromycin as described previously (*McKinley and Cheeseman, 2017*). Additional cell cycle and chromosome inducible knockouts were from *McKinley and Cheeseman, 2017*. For the Ki67 stable knockout cell line, the HeLa cell inducible knockout version was induced with Dox and subsequently sorted by FACS to create clonal cell lines, which were screened using immunofluorescence against Ki67.

## Single-molecule RNA fluorescence in-situ hybridization (smFISH)

Custom Stellaris RNA-FISH probes labeled with Quasar dyes (i.e., Quasar570 or Quasar670) were designed against specific centromere RNAs and purchased from Biosearch Technologies (Petaluma, CA) and PixelBiotech GmbH (Schriesheim, Germany). To conduct single-molecule FISH, cells were grown on poly-L lysine coverslip in 12-well plates were washed with PBS and fixed with 4% paraformaldehyde in 1X PBS containing RVC (Ribonucleoside Vanadyl Complex) for 10 min at room temperature (RT). After washing cells twice with 1X PBS, cells were permeabilized in 70% ethanol for at least 20 min at 4°C. Cells were pre-incubated with 2X SSC; 10% deionized formamide for 5 min, and incubated with hybridization mix (0.1 μM RNA-FISH, 10% deionized formamide, in Hybridization Buffer (Biosearch Technologies)) overnight at 37°C in the dark. Finally, cells were washed twice with 10% deionized formamide in 2X SSC for 30 min at 37°C and once with Wash B (Biosearch Technologies) for 5 min at RT. For experiments with immunofluorescence coupled to smFISH, HeLa cells grown on poly-L lysine coverslip in 12-well plates were washed with PBS and fixed with 4% paraformaldehyde in 1X PBS containing RVC (Ribonucleoside Vanadyl Complex) for 10 min at RT. After washing cells with 1X PBS, cells were permeabilized for 5 min at RT with 0.1% Triton-X in PBS with RVC. After washing, primary and secondary antibody incubation was performed at RT for 1 hr in PBS and RVC. Antibody concentrations: DM1a (anti-tubulin; Sigma): 1:10000, ACA (anti-centromere antibodies – human auto-immune serum; Antibodies, Inc): 1:1000, CENP-A (Abcam, ab13939) at 1:1000, Fibrillarin (Abcam, ab5821) at 1:300, and Ki67 (Abcam, ab15580) at 1:100. For smFISH, cells were fixed again for 10 min with 4% PFA in 1x PBS. Hybridization was performed as above. Coverslips were mounted on cells with Vectashield containing Hoechst.

For imaging, slides were imaged using a DeltaVision Core microscope (Applied Precision/GE Healthsciences) with a CoolSnap HQ2 CCD camera and 60x and 100 × 1.40 NA Olympus U- PlanApo objective. smFISH foci were counted per nucleus from z-projected images using CellProfiler (*Carpenter et al., 2006*). Prior to projection, each z-section was examined and the appropriate z-slices were projected (max intensity projection). For the analysis of smFISH foci, we used a projection of the entire cell volume. For the analysis of the overlap between smFISH foci in the nucleus and the presence of centromeres in the nucleolus, we analyzed individual z sections. Image files were processed using Deltavision software or Fiji (*Schindelin et al., 2012*).

## DNA FISH

Custom multiplexing single-molecule FISH (smFISH) HuluFISH probes labeled with Atto565 were designed against specific centromere RNAs and purchased from PixelBiotech GmbH (Schriesheim, Germany). To conduct DNA FISH, cells were grown on poly-L lysine coverslip in 12-well plates were washed with PBS and fixed with 4% paraformaldehyde in 1X PBS for 10 min at room temperature (RT). After washing the cells with 1X PBS for 10 min, cells were permeabilized in 70% ethanol at −20°C overnight. Cells were treated with RNase A (Qiagen) at 1:1000 for 30 min at 37°C followed by denaturation in 70% deionized formamide; 2X SSC buffer at 75°C. The samples were then dehydrated in series of cold ethanol washes (70%, 90% and 100%) for 2 min each and air dried. Cells were washed with Hulu Wash (PixelBiotech) twice for 10 min at room temperature and incubated with hybridization mix (0.5 μL HuluFISH probes in 1X HuluHyb solution) overnight at 30°C in the dark. Finally, cells were washed twice with Hulu Wash for 30 min at room temperature.

## Reverse transcription and quantitative real-time PCR

Cells were harvested 24 hr after BMH-21 addition, 5 hr after THZ1 treatment, or 4 days after doxycycline addition to induce the CENP-C knockout. RNA was purified using TRIzol reagent (Life Technologies) according to manufacturer's instructions. 2 μg of total RNA was used in the cDNA synthesis reaction with the Maxima First Strand cDNA Synthesis kit for RT-qPCR (Thermo Scientific). We used twice the recommended volume of dsDNase and allowed the DNase treatment to proceed for 30 min at 37°C. The increased concentration of dsDNase and length of reaction was critical for the complete removal of genomic DNA. The cDNA was subjected to quantitative real-time PCR using the SYBR green PCR mastermix (Thermo Scientific) according to manufacturer's protocol. Standard curves were used for quantitative assessment of RNA levels and centromeric RNA levels were normalized to GAPDH mRNA. If the levels of centromeric RNA fall far below of the linear range in the standard curve, we noted that the RNA was not detectable and was set to 0 in the figures. After

normalizing centromeric RNA levels to GAPDH, all of the data was normalized to HeLa. For RNA half-life experiments, HeLa or CENP-C inducible knockout cells (induced for 4 days) were treated with THZ1 and RNA was isolated the indicated times after THZ1 inhibition for RT-qPCR analysis as described above. The chromosome 21 alpha-satellite RNA levels were normalized to a stable mRNA, GAPDH. To calculate half-life the data was fit to an initial plateau followed by single exponential decay using GraphPad Prism. GAPDH primers: 5'-TCGGAGTCAACGGATTTGGT-3' and 5'-TTCCCG TTCTCAGCCTTGAC-3'. chromosome 21 (CH21) primers: 5'-GTCTACCTTTTATTTGAATTCCCG-3' and 5'-AGGGAATGTCTTCCCATAAAAACT-3' (*Nakano et al., 2003*; *Molina et al., 2016*).

## Acknowledgements

We thank the members of the Cheeseman lab and Gayathri Muthukumar for their support and input. This work was supported by grants from The Harold G and Leila Y Mathers Charitable Foundation, the NIH/National Institute of General Medical Sciences (R35GM126930) to IMC, and an American Cancer Society post-doctoral fellowship to LB. The authors declare that they have no conflict of interest.

## Additional information

### Funding

| Funder | Grant reference number | Author |
|---|---|---|
| National Institute of General Medical Sciences | R35GM126930 | Iain M Cheeseman |
| American Cancer Society | | Leah Bury |
| G. Harold and Leila Y. Mathers Foundation | | Iain M Cheeseman |

The funders had no role in study design, data collection and interpretation, or the decision to submit the work for publication.

### Author contributions

Leah Bury, Conceptualization, Funding acquisition, Investigation, Visualization, Methodology, Writing - review and editing; Brittania Moodie, Investigation, Visualization, Methodology, Writing - review and editing; Jimmy Ly, Conceptualization, Validation, Investigation, Visualization, Methodology, Writing - review and editing; Liliana S McKay, Investigation, Visualization; Karen HH Miga, Formal analysis, Methodology, Writing - review and editing; Iain M Cheeseman, Conceptualization, Formal analysis, Supervision, Funding acquisition, Visualization, Writing - original draft, Writing - review and editing

### Author ORCIDs

Liliana S McKay http://orcid.org/0000-0001-5426-5539
Iain M Cheeseman https://orcid.org/0000-0002-3829-5612

### Decision letter and Author response

Decision letter https://doi.org/10.7554/eLife.59770.sa1
Author response https://doi.org/10.7554/eLife.59770.sa2

## Additional files

### Supplementary files
- Supplementary file 1. Table showing sequences for smFISH probes.
- Supplementary file 2. Table showing analysis of matches of smFISH probe sequences to centromere reference sequences.
- Transparent reporting form

## Data availability

All data generated or analyzed during this study are included in the manuscript and supporting files.

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
