## [Decision Letter]

[Editors' note: this paper was reviewed by Review Commons.]

**Acceptance summary:**

Centromeres are needed for the proper inheritance of chromosomes and – although centromeres typically do not contain genes – they are being transcribed. These transcripts are important for centromere function, but their regulation and role is still poorly understood. Centromeres are often, but not always, located next to nucleoli, with which they have an intricate, but also incompletely understood, relationship. This paper now adds one more facet to this picture by showing that contacts between centromeres and the nucleolus influence the abundance of centromeric transcripts. This reinforces the relevance of the centromere-nucleolus interaction and raises the question how physiologic or pathologic disruptions of this interaction influence centromeres and chromosome segregation.

**Decision letter after peer review:**

Thank you for submitting your article "Alpha-satellite RNA transcripts are repressed by centromere-nucleolus associations" for consideration by *eLife*. We have considered the reviews from Review Commons as well as your response. Since one of the reviewers had a conflict of interest, we solicited additional advice. The comments below reflect the discussion between all involved. The evaluation has been overseen by Silke Hauf as Reviewing Editor and Anna Akhmanova as the Senior Editor.

As the editors have judged that your manuscript is of interest, but as described below that additional experiments are required before it is published, we would like to draw your attention to changes in our revision policy that we have made in response to COVID-19 (https://elifesciences.org/articles/57162). First, because many researchers have temporarily lost access to the labs, we will give authors as much time as they need to submit revised manuscripts. We are also offering, if you choose, to post the manuscript to bioRxiv (if it is not already there) along with this decision letter and a formal designation that the manuscript is "in revision at *eLife*". Please let us know if you would like to pursue this option.

Summary:

Transcription of centromeric DNA is still poorly understood. It remains unclear or controversial how, where and when centromeric transcripts are produced, how long they are, how they are modified, where they are located, how their levels are controlled, and what their functions are. Here, you used single-cell smFISH to localize and quantify centromeric transcripts in human cell lines. You confirm previous reports that RNA Polymerase II is required for the production of centromeric transcripts. In contrast, you find that a large number of centromere proteins, kinetochore proteins or chromatin regulators are not required for centromere transcription. On the contrary, depletion of the inner kinetochore protein CENP-C increased the number of transcripts. (The reason for this remains unclear.)

Unlike prior reports, your data suggest that centromeric transcripts do not remain associated with centromeres, although they remain in the nucleus until nuclear envelope breakdown in mitosis. Furthermore, you find a negative correlation between the number of centromere transcript foci and the fraction of centromeres that co-localize with nucleoli, and you interpret this as evidence that centromere-nucleolus interactions may limit centromeric transcription.

Overall, your data adds to the diverse picture of findings on centromeric transcripts, and will surely inspire future investigations into this topic.

Several conclusions that you want to make require additional experimental support:

1) All reviewers indicated that controls are needed for the iKO experiments. Although you have validated these cell lines previously, the controls are required to demonstrate that knock-outs/knock-downs occurred in your current experiments, ideally in the cells that you are analyzing. If the antibodies do not work in the smFISH experiment, you could provide immunoblots. Minimally, this should be shown for the particularly relevant CENP-C iKO cells.

If you do not have antibodies available, or in order to get an estimate for the fraction of cells with efficient knock-down, a quantification of cellular phenotypes in the population could also be useful. According to your reply, you have already performed the latter. Please provide these data.

2) In all conditions where you observe a larger number of smFISH foci, you interpret this as an increase in transcription. Before making this conclusion, alternative reasons need to be excluded:

i) It needs to be checked whether the increased number of transcripts could reflect stabilization (increased half-life) rather than increased transcription. Two experiments seem important: Is the drastic drop in transcript levels during G1 still observed in the knock-outs/inhibitions that increase transcript levels? Is the half-life during S/G2 increased? (Although prior reports have described centromeric transcripts as very stable, the fact that you see drastic drops after 5 hours of RNA Pol II inhibition, suggests that the transcripts that you are assaying are turning-over at a detectable rate.)

ii) For the CENP-C iKO cells that show an increased number of ASAT transcripts, it needs to be addressed whether this can be attributed to failed cytokinesis. Is it possible that the cells have become larger / diploid, and that this is the sole reason for the higher number of transcripts? Similar concerns could apply to some of the other knock-outs tested. A quantification of transcripts and centromeres in the same cells could be useful.

3) The conclusiveness of your results would greatly profit from confirmation by a different approach. Quantitative RT-PCR was suggested, and you already indicated it should be feasible to test for centromeric transcripts by qPCR in your key experimental conditions (CENP-C iKO, Ki67 iKO, fibrillarin iKO, RNA polymerase I inhibition, RNA polymerase II inhibition).

4) The conclusiveness of the results would also profit from showing that the FISH probes indeed detect alpha satellite RNA. We suggest that you perform a DNA-FISH experiment using the same probes.

---

## [Author Response]

Reviewer #1:In this manuscript the authors explore the requirements for centromere transcription using single-molecule FISH. Previous studies have found that centromeres are transcriptionally active in a wide variety of organisms. Centromere transcription has been proposed to facilitate Cenp-A deposition through chromatin remodeling and to directly contribute to centromere/kinetochore function by producing a functional ncRNA. However, we currently know almost nothing about how transcription is initiated at the centromere or how levels of centromere transcripts are controlled. This manuscript makes several major findings that are potentially of importance to groups studying centromere transcription. 1. Centromere RNAs are produced by RNA Polymerase II and are localized in the nucleus of a wide-range of cell types. 2. Centromere RNAs do not localize to the centromere, which is in contrast to several recent studies. 3. Centromere proteins are not required for transcription of alpha-satellite sequences. 4. Localization of centromeres to the nucleolus represses centromere transcription. Overall, this is a solid manuscript and has the potential to make a significant impact in the field. Below I suggest a couple of experiments and modification to the data presentation that could improve the manuscript.

We thank this reviewer for their interest in this paper and agree with their clear articulation of the key points.

1) All of the experiments in this manuscript rely on detection of centromere RNAs using single molecule FISH probes. These probes are validated by showing the RNase treatment removes the FISH signal. A strength of this approach is that the authors use multiple different probe sets and achieve comparable results. However, there is no orthogonal validation that the probes detect alpha satellite RNA. All of the experiments in this manuscript would be significantly improved by showing that the results presented here can be confirmed by a different approach. I suggest that the authors use Q-RT-PCR to validate the smFISH results.

The smFISH probes provide a powerful and unique strategy to detect alpha-satellite transcripts. To ensure that these experiments are carefully controlled, we analyzed multiple distinct probe sequences that recognize alpha-satellite transcripts derived from different chromosomes, as this reviewer highlights. We also conducted an in-depth computational analysis to ensure that these probes do not match genomic sequences outside of alpha-satellite regions. However, we recognize and agree that a complementary method to detect these transcripts would be a useful addition to this paper. We are currently highly constrained in our ability to conduct these experiments due to COVID-19-related laboratory closures, but if feasible our goal for a revised manuscript would be to conduct qPCR experiments for a subset of the conditions that are the most central to the key results in this paper (focusing particularly on HeLa and Rpe1 control cell lines, CENP-C iKO, Ki67 KO, and RNA Polymerase I and RNA Pol II inhibitors).

2) Several results in this manuscript directly contradict results in published studies, but these discrepancies are not discussed. I believe the authors need to discuss the following discrepancies between their results and those in the literature:a) McNulty et al., 2017. Show that alpha-satellite RNA is transcribed from all centromeres and remains localized to the site of transcription. The different results and possible explanations for the differences should be discussed.b) Additionally, Rosic et al., 2014, Blower, 2016 and Bobkov et al., 2018 all show that centromere RNAs localize to centromere regions. The differences between these studies and the authors results should be discussed.c) The authors show that satellite RNA cannot be detected on mitotic chromosomes. However, Johnson et al., 2017, Bobkov et al., 2018, and Perea-Resa et al., 2020 show that EU-labeled RNA can be detected at the centromere during mitosis. The authors should discuss the discrepancy between their results and these studies. Is it possible that their smFISH probes do not detect nascent, chromatin-bound transcripts?

We believe that a strength of our paper is that it assesses alpha-satellite transcripts in individual intact cells using fixation conditions that preserve the native behaviors without disruptive and harsh extraction. As our results differ from those of other laboratories in some cases, we agree that it would be helpful to comment more directly on these differences with prior work. Points a, b, and c above all relate to the presence of alpha-satellite transcripts at centromeres. For the revised paper, we will include a discussion of these prior observations and some possible reasons for the differing results. In particular, we think that these discrepancies reflect two key differences:

1) Other strategies with harsh extraction conditions likely eliminate soluble alpha-satellite transcripts that are not tightly associated with centromeres, whereas our work preserves these.

2) It is possible that we are unable to detect nascent transcripts by smFISH as these are embedded within the RNA polymerase.

Extraction conditions: An advantage of the smFISH probes used in our paper is that these require mild fixation conditions without prior extraction to better preserve cellular structures allowing us to analyzed intact cells, rather than chromosome spreads. Thus, our approach maintains the diverse alpha-satellite transcripts that are not bound to centromeres, and which may have been washed away in other studies. In contrast, some prior studies used stringent extraction conditions and primarily conducted experiments in chromosome spreads (not intact cells). Although it is not feasible to precisely determine the basis for differences without repeating this work the precise approaches and conditions from each paper and working closely with each group, we believe that these substantial technical differences explain our differing observations that reveal that the majority of alpha-satellite transcripts do not remain at centromeres.

Nascent transcripts: As suggested by this reviewer, we agree that our differing conditions may mean that we are unable to detect nascent transcripts that are closely associated with the RNA polymerase, inaccessible due to their chromatin proximity, or that are not sufficiently elongated such that they are present to hybridize to multiple copies of the smFISH probes to be detectable. The alpha-satellite transcripts must be derived from centromeric and pericentromeric regions and so must exist there at some point (as also attested to the EU signals that this reviewer mentions in the work from our collaborative the Blower lab; we have also detected EU signal at centromeres). However, our work suggests that alpha-satellite transcripts do not persist at centromeres indefinitely once generated, with mature transcripts in the nucleoplasm and liberated from chromosomes during mitosis. We believe that the combination of the relative inability of our smFISH probes to detect nascent transcripts, but stringent conditions disrupting non-centromere bound transcripts for prior work likely explain these distinctions.

d) The authors show nicely that deletion of Ki-67 reduces centromere localization to the nucleolus and increases centromere transcription. However, this has no effect on centromere function. Studies from the Earnshaw lab (e.g. Nakano et al., 2008 and Bergmann et al., 2011) show that increasing or decreasing centromere transcription results in loss of kinetochore function on a human artificial chromosome. The authors should discuss the differences between their results and these studies. Is it possible that the small size of the HAC exaggerates the importance of the correct levels of centromere transcription?

We are big fans of the Earnshaw lab work. In this case, there are a couple of possibilities to explain the strong effect that the Earnshaw lab observed on kinetochore function by perturbing centromere transcription. First, the degree of the change in centromere transcription may make a big difference. The Ki-67 results in an approximately 2-fold increase in alpha-satellite smFISH foci, which may still be within a permissive range for normal kinetochore function. Second, the experiments from the Earnshaw lab rely on targeting activating or silencing proteins to the centromere region, and it is possible that changes in centromere chromatin downstream of these factors contribute to the observed phenotypes in addition to altering the amount of centromere transcription. We will include a brief discussion of the Earnshaw work in a revised paper.

3) The authors treat cells with transcriptional inhibitors for 24 hours. I am concerned that this may result in massive cell death. It would be helpful to include cell viability data from these experiments.

We appreciate this point and agree that cell lethality is an important consideration given the essential role of the RNA polymerases. For the inhibitors, we first treated the cells for a variety of different time points to evaluate these behaviors. For example, we found that we could treat cells with RNA Polymerase II inhibitors for as much 48-72 hours without detecting noticeable cell death. Thus, at the 24 hour time point, the cells remain viable and intact, as is also visible in the images showing DNA staining for these treatments in Figure 3. We also note that this timing is consistent with prior studies that block transcription or translation. However, we did additionally conduct these experiments at earlier time points (5 hours and 12 hours post-drug addition) and obtained similar results. For example, for the Cdk7 inhibitor using the ASAT probe, we observed the following smFISH foci/cell: Control (3.4 foci/cell), 5 h (1.5 foci/cell), 12 h (1.2 foci/cell), 24 h (0.9 foci/cell). There is a clear effect even at 5 hours of treatment and a continued downward trend. Both for simplicity and because the replicates and number of cells that were quantified were lower for these conditions, we chose not to include these in the paper. We will include a statement regarding these earlier time points in the revised version.

4) In Figure 3C the authors examine the effects of centromere protein knock outs on centromere transcription. To me this is the most important experiment in the manuscript and is a major step forward for the field. The authors use inducible CRISPR knock out cell lines that are not 100% penetrant. It would be helpful if the authors could describe how they ensured that cells included in the image quantification were knock out cells.

Based on this comment and the other questions from the other reviewers, we recognize that we need to provide a much better description of the CRISPR knockout strategy, the prior validation of these cell lines, and the strategies that allow us to use these cell lines in a robust manner to ensure that we are effectively eliminating the target genes. We have systematically tested this strategy in multiple cases and find that this strategy is superior to RNAi for its efficacy and the potency of the phenotype, particularly for this type of cell biological assay.

The Cas9-based strategy is a highly effective way to conditionally eliminate essential genes. In this case, the efficiency of the Cas9 nuclease ensures that the genomic locus is cleaved in essentially 100% of cases. As this is repaired in an error prone manner and typically using non-homologous end joining, 66% of individual events result in frame shifts mutations that disrupt the coding sequence of a target gene, with ~50% of cells resulting in frame shifts in both copies of a gene. In addition, if a sgRNA targets a region of a gene that cannot tolerate mis-sense mutations, this will result in an even greater fraction of mutant cells. Thus, these inducible knockout cell lines result in robust and irreversible gene knockout, with a large fraction of cells (50% or more) displaying a clear phenotype. However, it is also true that there are a subset of cells within the population that will repair the DNA damage following Cas9 cleavage in a way that preserves protein function such that they behave similarly to control cells. Importantly, this means that there will be two classes of cells within a population – those that are unaffected, and those that are strongly affected. As we are analyzing each cell individually instead of creating a population average, this will capture this phenotypic diversity to reveal two populations of behaviors in cases where eliminating a gene results in a substantial change in smFISH foci. For example, the smFISH foci/cell data for the CENP-C inducible knockout (Figure 3C and 3E) indicates that many cells have smFISH foci numbers that are comparable to control cells, but others that display substantial differences and highly increased numbers. An ideal control in these experiments would be to additionally analyze the levels of the target protein together with the smFISH analysis. Unfortunately, many of the antibodies are not compatible with the conditions needed for the smFISH. For CENP-C, the antibody that we have is not compatible with the conditions that we are using for the smFISH, so it is not feasible to co-stain these cells as suggested. Instead, for our analysis of the centromere-nucleoli localization (for example), we used the presence of a clear CENP-C interphase phenotype (“bag of grapes” resulting from chromosome mis-segregation) as an indication that the cells had been knocked out for CENP-C.

The majority of the Cas9-based inducible knockouts that we used for this paper were generated previously in the lab (McKinley et al., 2015; McKinley et al., 2017). For the centromere protein knockouts (McKinley et al., 2015), these were analyzed previously with respect to phenotype and monitored for the depletion of each gene target over time. For the larger collection of cell cycle and cell division inducible knockouts, for our prior work we systematically validated each of these with respect to their phenotype (see http://cellcycleknockouts.wi.mit.edu). Thus, we are confident that each of these cell lines is functional and effective for eliminating the target gene.

For conducting the experiments using the inducible Cas9 cell lines in this paper, we used the presence of these previously-defined phenotypes within the population as a validation that the strategy is working. Again, in general we find these knockouts are both penetrant and severe in their phenotypes. Importantly, for this diverse set of genes, we note that our goal was to broadly survey diverse factors to identify changes in alpha-satellite transcript levels. We intended this analysis as a “screen” where we would identify factors that resulted in a substantial change in the number of smFISH foci. As with any larger analysis, it is possible that there are false negatives where we did not detect a strong effect on transcript levels (such that they may contribute to centromere transcription). We have tried to use caution not to indicate that this data excludes any possible role for these factors in transcript levels, although in general the majority of the tested factors did not show a substantial change in smFISH foci. For the revised paper, we will make an explicit statement to this effect.

5) The authors cite Quenet and Dalal., 2014 for the idea that transcription during G1 is important for new Cenp-A loading. They should also cite Chen et al., 2015 and Bobkov et al., 2018.

Thank you for these helpful suggestions. We will update the text to incorporate these references.

Reviewer #2:The study by Bury et al. investigates the formation of two different types of alpha-satellite transcripts (ASAT, SF1 and 3) in different human cell lines. Using smFISH they find that during the cell cycle these centromeric transcripts don’t stay at the centromere and are found in the cytoplasm after mitosis. Using specific inhibitors, they find that transcription is dependent on RNAPII, but not on various centromere and kinetochore proteins taking advantage of an inducible CRISPR-depletion system that the lab had previously developed. Interestingly, they find that CENP-C, a major component of the centromere and previously characterised as an RNA-binding protein, negatively regulates alpha-satellite transcript levels. Another regulator for transcript levels appears to be centromere-nucleolus interactions (as also indicated in the title) acting to suppress expression of these non-coding RNAs.This is overall a really interesting study and indeed, transcription at the centromere is little understood at this point. Given the importance of the centromere the findings in this manuscript will be of high interest to both researchers in the field and a general audience. There are novel and interesting insights into centromeric transcripts but the study still requires some controls.

We appreciate this reviewer’s kind words and their clear description of our work.

1) The authors state that the majority of smFISH foci do not colocalise with centromeres in a combined IF/FISH experiment (some quantification and a % of that subpopulation should be given somewhere). This is a bit concerning but of course could also be true. It either means that alpha-satellite transcripts leave the centromere as suggested by the authors (although some should be visible at the centromeres during the act of transcription). Alternatively, a trivial explanation would be that there is a lot of unspecific staining, which can occur in FISH-experiments to varying degrees. The RNase treatment to control for the absence of potential DNA hybridization is convincing, but the FISH probe could also interact with non-centromeric cellular RNA. With the centromere localisation as a reference point gone, some control is needed to validate that the RNA-FISH signals are indeed recognising alpha-satellite RNA that emerged from centromeres. The authors could try competition experiments titrating unlabelled specific or unspecific DNA probes alongside their labelled specific FISH probe into their FISH experiment to see if they lose or maintain the signal and the number of foci. The specific RNA FISH probes could also be used in DNA FISH, to demonstrate they are working and recognising specific centromeres.

For understanding this behavior, we believe that an important feature of alpha-satellite transcripts is that they are relatively stable (protected from nucleases within the nucleus), but that their overall number is low, consistent with transcription of other non-coding regions across the genome. Thus, if a transcript were produced at centromeres, but subsequently diffuses away, only a small subset would be detectable at centromeres. In addition to our validation these probes using RNAse, we would like to highlight that we have analyzed multiple distinct sequences that recognize different subsets of alpha-satellite repeats. In each case, the observed behaviors are very similar. In addition, the nature of the oligo FISH method requires multiple individual probes to anneal to the same transcript such that a signal is only detected if a sufficient number of oligos bind to the same transcript. This makes nonspecific binding unlikely to contribute to a false signal. Finally, a subset of the perturbations that we tested that are relevant to centromere function (including the CENP-C inducible knockout) clearly affect the levels of these transcripts, supporting a centromere origin. The additional control experiments suggested by the reviewer could be useful, but are technically complex with their own caveats in interpretation and we do not feel that they would add substantially to the existing paper. Instead, as discussed in response to reviewer #1, point #1, we plan to validate key results described in the paper using qRT-PCR (if possible based on current experimental constraints in the lab associated with COVD-19).

As described above in response to reviewer #1, point #2, we also believe that some differences with prior work suggesting that alpha-satellite transcripts localize to centromeres may be due to stringent extraction conditions that eliminated non-centromere bound transcripts, while at the same time reflecting our inability to detect nascent transcripts. Quantifying “colocalization” within the nucleus is limited by the resolution in light microscopy, and we would prefer to use caution in defining which transcripts in our smFISH analysis overlap with centromeres. However, we believe that our work clearly highlights the fact that a general feature of mature alpha-satellite transcripts is that they localize throughout the nucleoplasm and are not strongly associated with mitotic chromosomes.

2) Apart from Figure 4, there is no analysis shown for statistical significance. This should be done for most if not all quantifications. Are indeed ASAT and antisense RNA Foci number not significantly different? The authors say that the levels of alpha-sat RNA in Rpe1 cells are not substantially different from other cell lines, but is it also not significant (Figure 1F)? In Figure 2D it is concluded that transcripts foci number are increased in S/G2 (from G1) and remain stable in mitosis, but it looks like there is an increase in mitosis. Again, it looks like the higher number of smFISH foci/Cell is significantly higher for both ASAT and SF1, so some statistical analysis would be required here.

For this paper, we quantified hundreds of cells for each condition, measuring the number of foci/cell in each case. Because of these large n’s, even relatively small differences between samples become statistically significant when tested using standard statistical comparisons (unpaired T test and one-way ANOVA test amongst others). For our experiments, every sample condition included an analysis of control cells, allowing us to compare the control condition to any perturbations on the same day. However, there is some variability between these different replicates, with the average number of ASAT smFISH foci/cell in HeLa cells ranging from 3.4 to 5.6. When compared relative to each other, a subset of these control samples will appear to be statistically different from each other despite the fact that this is not a substantial difference between replicates. Similarly, the majority of the tested inducible knockout cell lines are statistically different from control cells, even when the differences are relatively minor. Therefore, we have tried to use caution when applying the double-edged sword of statistics to these analyses. Instead, we have tried to consider differences with a “substantial magnitude” instead of “statistically significant” differences that may make modest, but statistically significant differences seem artificially more important. We believe that the graphs in which every data point is represented, together with listing the average number of foci/cell in each condition allow the reader to evaluate this data for themselves. Many of the trends that this reviewer highlights are indeed interesting comparisons to consider for future work.

3) Starting with the description of Figure 1E in the main text the paper equates foci count of smFISH per cell with RNA transcript levels. I'm not convinced that these are necessarily the same. You could have many weak foci or few very bright with the same amount of overall transcripts in both. The authors start out introducing smFISH as highly sensitive "for accurate characterisation of number.…of RNA transcripts". This suggests that foci intensity could be used as a read-out for transcript levels. It should be possible to measure individual intensity of the foci for a subset of images. Do foci intensity correlate or anti-correlate with foci numbers? Is the sum of the intensities of all the foci less variable than the foci number for an individual cell type?

Due to the repetitive nature of alpha-satellite sequences, an increased intensity of a smFISH foci could reflect either the close proximity of multiple separable transcripts, or a longer transcript with multiple binding sites for the smFISH probes. Because of this, throughout the paper, we have referred to these as “foci” instead of stating a specific transcript number. As part of the automated computational analysis of the smFISH images, we additional analyzed foci intensity. In general, these values were similar across a cell population and between various perturbations with the key results and findings consistent whether we measured foci number or overall foci intensity per cell. However, foci intensity can vary slightly across a coverslip (technical constraints, not biological differences), and thus we have focused on foci number as a more consistent metric that correlates with the production of alpha-satellite transcripts.

4) I really like the use of the inducible CRISPR system to remove various centromere factors. However, some validation would be required to show that the system is effective in removing the proteins of interest in these experiments. For instance it would be helpful to show in Figure 3D an additional panel with CENP-C staining. Also for a subset of factors, some antibody staining co-staining with the smFISH could be provided in the supplemental material.

We appreciate this point. However, we feel that the existing experiments appropriately consider the nature of the knockout. First, we primarily used Cas9-based inducible knockouts that were generated previously in the lab (McKinley et al., 2015 and McKinley et al., 2017). As these knockouts have been described previously and extensively validated with respect to phenotype (in every case; see http://cellcycleknockouts.wi.mit.edu for example) and antibody staining (in selected cases), we have not repeated this here for the diverse cell cycle knockouts used. In general, we find these knockouts are both penetrant and severe in their phenotypes. Given the broad number of knockouts that we tested, this is not feasible in every case. We also intended this analysis as a type of “screen” where we could validate any “hits” that were observed, and will use caution in our wording not to imply that a negative result is decisive.

The important exceptions to this are CENP-C (which we analyzed more closely) and Ki67 (for which both the inducible and stable knockouts were generated for this paper). For Ki67, the antibody staining is shown and we believe that this is clear. For CENP-C, the antibody that we have is unfortunately not compatible with the conditions that we are using for the smFISH, so it is not feasible to co-stain these cells as suggested. For the smFISH analysis in the inducible CENP-C knockout, we analyzed every single cell, including some cells that are likely to have intact CENP-C levels. Thus, if anything, the potent increase in smFISH foci underrepresents the dramatic effect of CENPC depletion. Based on our prior work (McKinley et al., 2015) we found that the CENP-C knockout results in a pervasive “bag of grapes” phenotype in which chromosomes mis-segregate during mitosis and are packaged into separable interphase nuclei. For the analysis of the nucleoli, we selected cells that displayed this clear phenotype (as shown in the figures).

5) Since none of the CRISPR iKO has a particular inhibiting phenotype it would be useful to include some positive control in the CRISPR experiment. Would it be possible to use a CRISPR iKO target that affect some factor of the transcription machinery (RNA Pol II or similar) to reduce transcript levels?

Generating additional Cas9 iKO cell lines is feasible, but would be time consuming. In this case, we are not convinced of the value of generating and validating these additional cell lines (particularly with the additional current constraints due to COVID-19). For evaluating the role of the RNA polymerases, we believe that the effect of the drug treatment is clear. For creating a positive control to assess whether the CRISPR iKO strategy is a feasible way to conduct these experiments, we would like to highlight the CENP-C iKO cell line, which has a potent effect in this assay.

6) The authors find a negative correlation between the nucleolus-centromere association and the number of alpha sat foci. This is really interesting and they suggest that the nucleolus association could negatively regulate centromere transcription. However, this correlation is rather indirect in the sense that cells with a higher-degree of nucleolus-centromere localisation have fewer smFISH foci and the inverse, disruption of the nucleolus increases smFISH foci number as a whole. A model based on physical association would suggest that a nucleolus associated centromere produces less or no transcripts. Given that this is not a population-based assay, it should be possible to address this directly by analysing the location of individual centromeres and corresponding transcripts to strengthen the hypothesis. This could be done by either analysing the smaller subset of centromere-associated foci that colocalise with the smFISH signal and test whether the majority of these signals are proximal or distal to the nucleolus (this would not work or be less meaningful if the subpopulation is very small). Or doing a combined DNA/RNA FISH experiment. The expectation would be that DNA FISH signals of centromeres close to the nucleolus would not produce an RNA FISH signal somewhere else, and vice versa.

We predict that centromere-nucleolar associations are dynamic. Thus, we anticipate that centromeres would be associated transiently with the nucleolus (perhaps for a few hours), and that a given centromere would not be associated with the nucleolus in every cell at a specific time point. Thus, we believe that analyzing these behaviors across a diverse range of cells, as we did for this paper, is appropriate. In addition, technical considerations make these suggested experiments prohibitive. Defining the relationship between a centromere RNA and its originating centromere would require combined DNA and RNA FISH. The repetitive nature of alpha-satellite repeats and the strong similarity of these sequences between chromosomes makes it highly complex to visualize an individual centromere. Even if we were able to do this, the conditions required to simultaneously detect nucleoli (immunofluorescence), RNA (smFISH), and DNA (requires denaturation and hybridization) make this such that it would be complex to correlate the localization of an individual centromere with the levels of the corresponding alpha-satellite transcripts. In addition, these RNAs are likely to persist for an extended duration (possibly throughout the course of an entire cell cycle), such that they would not necessarily correlate with the current localization behavior of the centromere from which they are derived. For future work (beyond the scope of this paper), we plan to create cell lines expressing both centromere (CENP-A) and nucleolar markers (for example, Ki67) to conduct time lapse imaging to assess the dynamic associations between these structures.

7) At the end of the Abstract, the authors conclude that the control of centromere transcription might be regulated by the centromere-nucleolar contacts to modulate chromatin dynamics. What does that really mean? One possibility they give in the discussion is rejuvenating centromeric chromatin. It would be nice if they could show some effect along those lines at the centromere in one of the manipulations they did (either through inhibiting or increase transcription). At least as discussed in the paper (Figure 3—figure supplement 1) it appears that overall levels of CENP-A are not affected. Is this different for newly loaded CENP-A? Or some other aspect of chromatin dynamics that is modulated? I realise that this might have been difficult to detect and therefore missing in the current study.

In a separate study from our lab as part of our recent work (Swartz et al., 2018), we found that CENP-A is gradually incorporated at centromeres in non-dividing quiescent cells, including non-transformed human Rpe1 cells and starfish oocytes. In the case of oocytes, which contain a substantial pool of mRNAs such that they do not require ongoing transcription for viability, we found that inhibiting RNA Polymerase II and preventing ongoing transcription blocked the incorporation of newly synthesized histones, including both canonical histone H3 and CENP-A. We realize that our description of this prior work was not sufficient to understand our integrated model, which relies on information from both papers. For the revised paper, we will update our discussion to better describe this data and present our model.

8) The authors state that as cells entered mitosis, dissociation of smFISH foci from chromatin was observed. While the absence of co-localisation of DAPI and smFISH signals is obvious in mitotic cells, what evidence is there that smFISH foci are chromatin associated in interphase nuclei? Rephrasing this bit might avoid confusion here.

We appreciate this point. We did not mean to imply that the smFISH foci are bound to (or associate with) chromatin in the interphase nucleus. We will reword this as suggested.

Reviewer #3:The manuscript of Bury et al. addresses how alpha-satellite transcription around centromeres is regulated. Using smFISH to detect alpha-satellite RNA transcripts, the authors find that alpha satellites are transcribed by RNA pol II, but their transcription is independent of centromeric proteins. In addition, they present evidence that nucleolar association represses alpha-satellite transcription. The data is convincing, solid and generally supports the conclusions. The manuscript includes appropriate control experiments, such as test for the validity of the RNA FISH probes. The manuscript is well-written and easy to follow, also for someone who is not directly an expert in the field.The authors use a single-cell technique (smFISH) to look at the localization and transcription of alpha-satellite transcription from centromeres. The technical advance of this paper is limited, as smFISH is a well-established technique by now. Nevertheless, applying this single-cell approach to these repetitive regions has resulted in new insights regarding the regulation of alpha-satellite transcription, especially their localization of centromeres to nucleoli. Regarding the significance of these insights in the context of centromere biology/regulation and its literature is hard to evaluate for me, because this is not my field of expertise (my background is in single-cell transcription regulation). As a researcher from a related research field, I think the findings of this manuscript are mostly relevant for the direct research community of centromere and alpha-satellite biology, but not for researchers outside the field.

We appreciate these comments regarding the carefully controlled nature of our paper and the value of the advances for understanding alpha-satellite transcription. We also agree that smFISH is an established technique, although it has not been applied to these repetitive alpha-satellite sequences in prior work, allowing us to make important new observations using the studies in this paper.

1) The description of the inducible knock out cell lines is very limited. My main concern is how is checked that the gene is actually knocked out. I went back to the referenced paper, but it is still is not clear to me whether the new knockouts are sufficiently checked. It would be more convincing if the authors could show western blots or other evidence that their knockouts are working. In any case, the description of the knockout generation should be more elaborate.

This important point was also noted by the other reviewers. Please see our responses to reviewer #1 point 4 and reviewer #2 point 4. As described above, for a revised paper, we will provide an improved description of these knockout cell lines, our validation of these tools, and how we conducted the experiments in this paper.

2) The authors nicely show that there is an inverse correlation between nucleolar association of the centromere and alpha-satellite transcription. The data supports this claim, but given the many knockouts and cell lines that were tested, with many intermediate phenotypes (such as CENP-B), I find the correlation based on 4 points a bit sparse. I would recommend filling up Figure 4C with a few more mutants, to show that the inverse correlation holds for all mutants. These experiments would be straightforward for the authors, as the knockout/cell lines and techniques are already available.

We see a compelling general correlation between the fraction of nucleolar-localized centromeres and alpha-satellite transcript levels. Our goal for Figure 4C was to highlight this correlation for a selected subset of conditions. However, we do not believe that there will be a precise linear correlation between transcript levels and nucleolar centromeres under every condition. Indeed, it is quite possible that some perturbations would affect transcript levels without altering nucleolar associations. This is particularly true for perturbations that cause subtle phenotypes. Systematically analyzing centromere-nucleolar co-localization for each of the knockouts represents a substantial undertaking that we do not feel would contribute substantially to this existing paper.

3) The nucleolar repression is also supported by the Fibrillarin and Ki67 knockout. These are nice experiments which support their findings. What I am missing is whether these data quantitatively agree with the inverse correlation. Are these mutants completely lacking nucleoli, and if so, would you not expect both mutants to show the same upregulation? Similar to my point above, where do these mutants fall in the graph of Figure 4C?

For the perturbations described in this paper, we believe that inhibiting RNA Polymerase I most closely approximates the condition where nucleolar function is eliminated. Although Ki67 is a nucleolar protein in interphase, loss of Ki67 does not cause lethality indicating that nucleolar function is largely intact. We agree that it would be a good experiment to assess nucleolar-centromere associations in the Ki67 knockout. In fact, we have tried these experiments several times. However, due to the absence of Ki67 (for which we have the best localization tools), we instead needed to use Fibrillarin to monitor nucleoli. We have found this antibody to be much more finicky and not as readily compatible with the fixation conditions needed to detect centromeres. Thus far, we have not been able to generate clear data for this behavior.

4) Related to this, since their imaging techniques have single-cell resolution, I wonder if cells that contain many centromeres in the nucleolus have less alpha satellite transcripts than cell with few centromeres.

The correlation between centromere-nucleolar associations and alpha-satellite transcript numbers is strongly supported by our data across a population. However, analyzing this in individual cells is additionally complicated by the fact that we found that transcript levels vary over the cell cycle (low in G1, higher in S/G2). In addition, monitoring each of these markers in individual cells is technically complicated. Thus, while we appreciate this suggestion, we believe that our data stands on its own.

5) One claim that is a bit speculative is the suggestion that transcription itself and not the RNA may be required for the function of the alpha-satellites. This is indeed supported by the fact that most transcripts are not localized at the centromeres. However, this contrasts to the findings of the papers that increasing alpha-satellite transcription in different mutants does not appear to result in any phenotype on centromere function. For a non-expert, the function of these transcripts/transcription itself is not clear from the current manuscript, so I would recommend discussing the nuances of its functions in more detail in the discussion.

We agree that our model is speculative, but have chosen to include this to provide our perspective on the possible roles for centromere transcription based on this paper and our other recent work (Swartz et al., 2018). We believe that our data provide a context and set of constraints for potential roles of centromere transcription, but also agree that future work is needed to resolve these. Based on this comment and those from the other reviewers, we will also provide a better description of the data in the Swartz et al. paper, which analyzed different features of centromere transcription.

6) To quantify the smFISH data, the authors count the number of foci. From the images, it looks like the different foci have very different intensities. This may occur if the transcripts are different length when transcribed from different genomic regions. However, this may also occur if several RNA co-localize to the same spot, i.e. if one spot contains several RNAs. Can the authors verify that the distribution of spot intensities matches the expected intensities based on the different transcribed alpha-satellite regions?

Please see our response to reviewer #2, point #3.

[Editors' note: further revisions were suggested prior to acceptance, as described below.]

Several conclusions that you want to make require additional experimental support:1) All reviewers indicated that controls are needed for the iKO experiments. Although you have validated these cell lines previously, the controls are required to demonstrate that knock-outs/knock-downs occurred in your current experiments, ideally in the cells that you are analyzing. If the antibodies do not work in the smFISH experiment, you could provide immunoblots. Minimally, this should be shown for the particularly relevant CENP-C iKO cells.If you do not have antibodies available, or in order to get an estimate for the fraction of cells with efficient knock-down, a quantification of cellular phenotypes in the population could also be useful. According to your reply, you have already performed the latter. Please provide these data.

We have now performed immunofluorescence with antibodies specific to the gene target for the CENP-A, CENP-B, and CENP-C iKO cell lines (as representative knockouts that are particularly relevant to the conclusions in the paper). These data are included in Figure 4—figure supplement 1A. Our results indicate that, after induction with doxycycline, the localization of the target protein is substantially diminished or undetectable in the vast majority of cells in the population. In addition, the substantial chromosome mis-segregation that occurs following CENP-C knockout is clearly visible based on the misshapen interphase nuclei. We have also already included immunofluorescence data for the stable Ki67 knockout (Figure 5—figure supplement 1D). In combination with our previous validation of the inducible knockout cell lines (McKinley et al., 2017), we believe that this strongly validates this approach.

We also note that the majority of the Cas9-based inducible knockouts that we used for this paper were generated and validated previously (McKinley et al., 2015; McKinley et al., 2017). For the centromere protein knockouts (McKinley et al., 2015), these were analyzed previously with respect to phenotype and monitored for the depletion of each gene target over time. For the larger collection of cell cycle and cell division inducible knockouts, for our prior work we systematically validated each of these with respect to their phenotype (see http://cellcycleknockouts.wi.mit.edu). Thus, we are confident that each of these cell lines is functional and effective for eliminating the target gene. However, as these other knockouts are not a focus of the paper and we do not have specific antibodies in each case, we have refrained from reanalyzing these for the broad collection of gene targets tested in Figure 4A and Figure 4—figure supplement 1B. Based on our extended prior work, we find this Cas9 inducible knockout strategy to be both penetrant and severe in their phenotypes.

2) In all conditions where you observe a larger number of smFISH foci, you interpret this as an increase in transcription. Before making this conclusion, alternative reasons need to be excluded:i) It needs to be checked whether the increased number of transcripts could reflect stabilization (increased half-life) rather than increased transcription. Two experiments seem important: Is the drastic drop in transcript levels during G1 still observed in the knock-outs/inhibitions that increase transcript levels? Is the half-life during S/G2 increased? (Although prior reports have described centromeric transcripts as very stable, the fact that you see drastic drops after 5 hours of RNA Pol II inhibition, suggests that the transcripts that you are assaying are turning-over at a detectable rate.)

We agree that it is important to consider differences in RNA half-life in addition to changes in transcription to account for differences in transcript levels. For the revised paper, we have now measured the alpha-satellite RNA half-life in control HeLa cells and in the CENP-C inducible knockout. For these experiments, we inhibited the synthesis of new RNA by RNA Pol II inhibition (THZ1 treatment to inhibit Cdk7) and measured alpha-satellite RNA levels over time by RT-qPCR. We found that in both control cells and CENP-C depleted cell, the half-life of centromeric RNAs is ~70 minutes (Figure 4H) with the transcripts substantially reduced by 5 hours of treatment. We believe that these data provide a strong addition to the conclusions of the paper and suggest that the changes in transcript number reflect changes to the level of transcription instead of altered transcript stability. Although we believe that the CENP-C iKO data provides a valuable addition, we chose not to test RNA half-life as part of the cell cycle analysis due to technical complications and the fact that these inhibitor treatments may alter cell cycle state or progression. Together, these results combined with the smFISH data strongly suggest that the observed changes occur through a transcriptional, rather than post-transcriptional, effect on alpha-satellite RNA levels.

ii) For the CENP-C iKO cells that show an increased number of ASAT transcripts, it needs to be addressed whether this can be attributed to failed cytokinesis. Is it possible that the cells have become larger / diploid, and that this is the sole reason for the higher number of transcripts? Similar concerns could apply to some of the other knock-outs tested. A quantification of transcripts and centromeres in the same cells could be useful.

For this revised paper, we have now assessed key conditions, such as the CENP-C inducible KO, using RT-qPCR for an alpha-satellite array associated with chromosome 21. We have normalized alpha-satellite transcript levels to that of GAPDH, which allows us to assess changes in transcription relative to cell size and other features. Importantly, our qPCR analysis reveals very similar behaviors as our smFISH analysis. In addition, we note that, based on our analysis of the CENP-C knockout phenotype (for both prior work and this paper), we do not detect a substantial increase in nuclear size, DNA intensity, or centromere numbers that would indicate the presence of a failure in cytokinesis. Instead, we detect misshapen nuclei that are consistent with cell division occurring with substantial chromosome mis-segregation.

3) The conclusiveness of your results would greatly profit from confirmation by a different approach. Quantitative RT-PCR was suggested, and you already indicated it should be feasible to test for centromeric transcripts by qPCR in your key experimental conditions (CENP-C iKO, Ki67 iKO, fibrillarin iKO, RNA polymerase I inhibition, RNA polymerase II inhibition).

For this revised paper, we have invested substantial effort to optimize conditions for RT-qPCR to detect centromere transcripts, as well as test key conditions that are relevant to the paper. We now present data for the qPCR for a previously published primer pair that is specific to Chromosome 21 (Nakano et al., 2003). This data is included in Figures 1F, 3D, 4E, and Figure 5—figure supplement 1E for HeLa cells, Rpe1 cells, RNA polymerase I inhibition, RNA polymerase II inhibition, CENP-C iKO, and Ki67 KO. We are excited for this data, which we believe substantially add to the paper and the strength of the conclusions.

Overall, our data indicate highly similar findings for the qPCR data as compared to our previous smFISH data. We find that Rpe1 cells display reduced transcript levels as compared to HeLa cells (Figure 1F), that the abundance of these transcripts depends upon RNA Polymerase II activity (Figure 3D and 4H), and that transcript levels are substantially increased following the inhibition of RNA Polymerase I (Figure 3D), and that transcript levels increase when CENP-C is eliminated (Figure 4E). For these conditions that display substantial changes in transcript levels, the results are striking. In contrast, we do not detect a dramatic change in alpha satellite transcript levels by qPCR for the Ki67 knockout (Figure 5—figure supplement 1E). We have noted this difference in the text, which may reflect the degree of sensitivity of each assay, technical differences, single molecule vs bulk assays, or a less potent role for Ki67.

4) The conclusiveness of the results would also profit from showing that the FISH probes indeed detect alpha satellite RNA. We suggest that you perform a DNA-FISH experiment using the same probes.

For our previous smFISH probes, the versions that we obtained from the manufacturer (Biosearch Technologies) were not suitable for DNA-FISH experiments. We have now obtained an additional set of probes using identical sequences, but from a second manufacturer (PixelBiotech GmbH). Using the RNA FISH protocol, we validated these probes and found that they displayed identical behavior for the detecting alpha satellite transcript levels (including number of smFISH foci, the sensitivity to RNAse A treatment, and the localization of the smFISH foci). We then used these probes in a modified protocol to conduct DNA-FISH (requires harsher conditions to denature the DNA). Under these conditions, these probes now display multiple clear foci that are consistent with centromeres – i.e., they localize throughout the nucleus in interphase and align during mitosis in the same way as a centromere marker would (similar to the staining of centromere components that we have been conducting for years). Although we are unable to co-localize these foci due to the fixation and extraction conditions, we believe that these images provide additional support for the specificity of these probes. These data are included in Figure 1—figure supplement 1A.